# VISIONSELECTOR: END-TO-END LEARNABLE VISUAL TOKEN COMPRESSION FOR EFFICIENT MULTIMODAL LLMS

## ABSTRACT

Multimodal Large Language Models (MLLMs) encounter significant computational and memory bottlenecks from the massive number of visual tokens generated by high-resolution images or multi-image inputs. Previous token compression techniques are often constrained by heuristic rules that risk discarding critical information. They may suffer from biases, such as attention sinks, that lead to sharp performance drops under aggressive compression ratios. To address these limitations, we reformulate token compression as a lightweight plug-and-play framework that reformulates token compression into an end-to-end learnable decision process. To be specific, we propose VisionSelector, a scorer module decoupled from the MLLM backbone that incorporates a differentiable Top-K mechanism and a curriculum annealing strategy to bridge the training–inference gap, enabling efficient and adaptive token selection various arbitrary compression rates. Remarkably lightweight with only 12.85M trainable parameters, VisionSelector demonstrates generalization across various compression rates and adaptively identifying critical tokens. This leads to superior performance across all compression budgets, evidenced by preserving 100% accuracy on MME with 30% retention budget, outperforming prior methods by 12.14% at 10% retention budget, and doubling prefill speed. Code and models will be publicly available.

## 1 INTRODUCTION

Multimodal Large Language Models (MLLMs) (Liu et al., 2023; Bai et al., 2025; Wang et al., 2025b) exhibit remarkable capabilities in complex vision-language tasks. However, their superior performance hinges on effectively handling high-information-density visual inputs, such as high-resolution images, multi-image sequences, and videos. This inevitably results in an explosion of visual tokens, exacerbating computational and memory bottlenecks during training and inference. Recent studies reveal significant redundancy in visual information, suggesting that effective token compression (Wang et al., 2025c; Shao et al., 2025b) could alleviate these bottlenecks. Consequently, visual token compression has emerged as a critical research frontier in MLLMs, aiming to enhance efficiency while preserving critical information.

Current token compression methods for vision models exhibit inherent structural trade-offs that constrain their effectiveness. Transformation-based approaches (Li et al., 2025), while preserving structural cues, enforce fixed compression ratios, lack flexibility, and demand separate training. Similarity-based methods (Wen et al., 2025; Alvar et al., 2025) prioritize token diversity but often discard fine-grained, task-critical signals, compromising performance. Attention-based techniques (Chen et al., 2024; Yang et al., 2025b), though intuitive, are vulnerable to biases such as attention sink (Zhang et al., 2024b; Xiao et al., 2023; Yang et al., 2025b; Weng et al., 2024), which can significantly degrade performance under aggressive compression. Compounding these issues, the reliance on model-specific feature statistics limits generalization (Shao et al., 2025b), underscoring the need for a more robust, adaptable compression framework.

To overcome these limitations, we propose a paradigm shift from heuristic-based post-processing to an end-to-end, optimization-driven token selection process. Our approach, VisionSelector, addresses the limitations of prior methods by introducing a lightweight, plug-and-play framework that

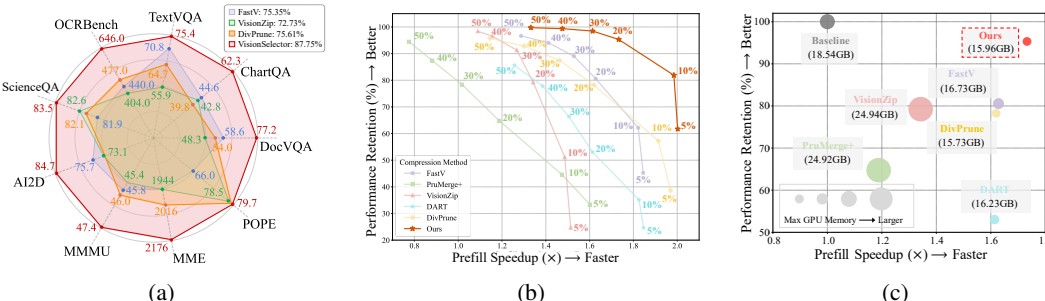

(a)                    (b)                    (c)

Figure 1: **Performance and Efficiency of VisionSelector.** **(a)** VisionSelector outperforms prior SOTA under a $10\%$ token budget on Qwen2.5-VL-7B, retaining $87.75\%$ performance on average. **(b)** Performance–speedup trade-off on DocVQA across 6 retention budgets. VisionSelector is optimal across all ratios, achieving a $2\times$ prefill speedup with only $5\%$ tokens. **(c)** DocVQA with a $20\%$ token budget: comparison of accuracy, GPU memory, and speedup. VisionSelector delivers a leading three-way trade-off.

seamlessly integrates with existing vision-language models and supports adaptive compression. VisionSelector comprises three key components: (1) a Differentiable Top-K Selection Mechanism that employs continuous relaxation to preserve end-to-end gradients while maintaining compatibility with high-performance kernels such as FlashAttention (Dao et al., 2022; Dao, 2023); (2) a Curriculum Annealing Strategy paired with the composite loss, which bridges the train–inference gap by progressively transitioning from soft to hard selection for robust importance learning; and (3) a backbone-decoupled Learnable Importance Scorer (LIS) that computes global token importance within a single forward pass, enabling a model trained at a fixed compression rate to generalize to various compression budgets at inference time. Requiring only 12.85M trainable parameters and approximately 40 minutes of training on 8 A800 NVIDIA GPUs, VisionSelector is both lightweight and cost-efficient. As shown in Figure 1, VisionSelector delivers substantial advancements. Specifically, at a $10\%$ token retention, it improves overall performance by 12.14 percentage points. At $20\%$ retention, it accelerates the prefill phase by a factor of 1.73× while simultaneously reducing memory consumption to $86.08\%$. The contributions of this paper could be summarized as:

- We propose VisionSelector, a novel framework that reformulates visual token compression from heuristic-based post-processing to an end-to-end learnable decision process, optimized directly by downstream losses.

- We design a lightweight, plug-and-play, and training-efficient module (12.85M parameters) that seamlessly integrates with existing MLLMs. It requires no backbone modification, maintains compatibility with various acceleration techniques(e.g., FlashAttention), and its learned mechanism effectively alleviates heuristic-driven biases such as sink attention.

- Trained at a single fixed compression rate, our approach demonstrates exceptional adaptability by generalizing to various compression budgets during inference. Our method significantly reduces memory usage and prefill time while maintaining superior accuracy, outperforming baselines across various compression budgets on 13 image and video understanding benchmark.

## 2 RELATED WORK

Inputs such as high-resolution images, multi-image sequences, and videos lead to a rapid surge in visual tokens within Multimodal Large Language Models (MLLMs). This expansion imposes substantial computational and memory burdens on Transformer-based architectures (Shao et al., 2025b). Consequently, visual token compression is critical for the efficient deployment of MLLMs (Wan et al., 2024; Tu et al., 2024). Existing compression methods generally fall into two main categories which are training-free approaches and learning-based algorithms.

## 2.1 TOKEN COMPRESSION METHOD

**Training-free token compression method.** Existing training-free token compression methods rely primarily on heuristic rules and categorize generally into attention-based approaches (Chen et al., 2024; Shang et al., 2024; Arif et al., 2025; Yang et al., 2025b; Zhang et al.; Shao et al., 2025a) and similarity-based techniques (Bolya et al.; Alvar et al., 2025; Wen et al., 2025; Wang et al., 2025a; Yang et al., 2025a; Tan et al., 2025; Sun et al., 2025). Attention-based strategies achieve compression by discarding tokens that possess low attention scores. For instance, FastV (Chen et al., 2024) utilizes visual-text attention scores. TokenCarve (Tan et al., 2025) advances this by incorporating the rank of the attention output matrix to quantify and preserve information-rich tokens alongside attention scores. However, these methods suffer from performance degradation due to the attention sink phenomenon (Wang et al., 2025c) and attention dispersion (Zhang et al., 2024b). Conversely, similarity-based approaches compress sequences by clustering or merging similar tokens. DART (Wen et al., 2025) identifies groups of repetitive tokens with high similarity for compression whereas DivPrune (Alvar et al., 2025) selects tokens by maintaining semantic diversity. Nevertheless, similarity-based methods risk losing critical fine-grained information (Shao et al., 2025b). Furthermore, heuristic-based methods rely heavily on the internal feature distribution of specific models and thus exhibit limited generalization across different architectures and scales.

**Learning-based token compression method.** These approaches explicitly train auxiliary modules to enhance compression and generally fall into three categories. **Transformation-based** methods reduce token counts or dimensionality via structured operations like pixel-shuffle (Wang et al., 2025b), convolution (Wei et al., 2025), resampling (Li et al., 2025; Liu et al., 2024b) or MLP projection (Bai et al., 2025), while **query-based** methods distill tokens into fixed representations using learnable queries (Li et al., 2023b; Ye et al., 2024; Alayrac et al., 2022; Ye et al., 2025; Zhang et al., 2025). Both strategies suffer from fixed compression ratios and altered feature distributions, often necessitating expensive retraining to maintain performance. In contrast, **token selection-based** methods identify informative tokens through lightweight scoring, preserving original features for dynamic inference. However, existing techniques (Huang et al.; Jiang et al., 2025) rely on local binary decisions rather than global ranking, causing sensitivity to compression rate variations. To address this, we propose VisionSelector, a general end-to-end framework that treats MLLM visual token selection as a discrete optimization problem, shifting the paradigm from local independent gating to global differentiable ranking.

## 2.2 DIFFERENTIABLE SORTING OPERATOR

Differentiable sorting operator (Grover et al., 2019; Prillo & Eisenschlos, 2020; Petersen et al., 2022a) relaxes discrete permutations to enable end-to-end optimization. Early methods like NeuralSort (Grover et al., 2019) and SoftSort (Prillo & Eisenschlos, 2020) construct soft permutation matrices based on pairwise comparisons or distances. However, these approaches involve $O(N^2)$ complexity and suffer from optimization instability due to strict normalization constraints. Furthermore, they lack precise control over the active token count which causes a sparsity gap between training and inference. In contrast, the Differentiable Top-K operator (Zhu et al., 2025) directly approximates the selection process with linear time complexity $O(N)$. By avoiding full permutation matrix construction, it mitigates gradient conflicts and balances smoothness with precision via temperature adjustment. We therefore adopt this operator in VisionSelector to ensure stable gradient propagation during training and accurate hard selection during inference.

## 3 METHOD

To overcome the limitations of current visual token compression techniques, we propose VisionSelector, a framework that recasts token compression as an end-to-end, task-oriented, and learnable decision process. The framework is built upon three key components: a lightweight Learnable Importance Scorer (LIS) for evaluating global token significance, a Differentiable Top-K Selection (DTS) mechanism that enables gradient backpropagation, and a Curriculum Annealing Strategy (CAS) to align training and inference. This integrated design achieves efficient, flexible token compression while maintaining full compatibility with modern acceleration libraries like FlashAttention.

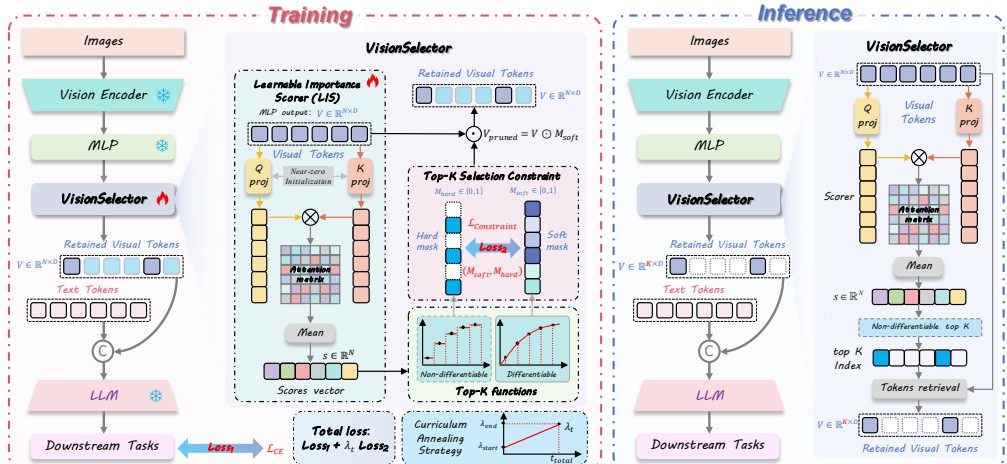

Figure 2: **Overview of our VisionSelector.** The framework introduces a lightweight, learnable importance scorer to evaluate token importance. During training, the Differentiable Top-K Selection produces a soft mask for gradient propagation, while a constraint loss integrated with a Curriculum Annealing Strategy progressively transforms it into a hard mask. At inference, the standard Top-K operation is applied for efficient token selection.

### 3.1 OVERALL FRAMEWORK

Our method, VisionSelector, is designed as a plug-and-play solution for seamless integration into advanced MLLMs such as Qwen2.5-VL (Bai et al., 2025). We deploy it between the modality interface and the large language model. The overall process, as illustrated in Figure 2, can be divided into the following steps: First, the modality interface projects the features from the visual encoder into visual tokens, $V \in \mathbb{R}^{N \times D}$, required by the LLM, where $N$ is the number of tokens and $D$ is the feature dimension. This set of features, $V$, is then fed into our proposed LIS, which generates an importance score for each token, yielding a score vector $s \in \mathbb{R}^N$. Based on a predefined compression budget $b$, the DTS utilizes the score vector $s$ to generate a soft mask, $M_{soft} \in [0,1]^N$. This soft mask is applied to the original visual token features via element-wise multiplication to suppress the expression of non-critical tokens: $V_{pruned} = M_{soft} \odot V$. Finally, the pruned visual features, $V_{pruned}$, are concatenated with the text token embeddings and fed into the LLM for subsequent processing. The entire model is trained end-to-end using the loss from the downstream task and a constraint loss designed to optimize the selection process.

### 3.2 LEARNABLE IMPORTANCE SCORER

As illustrated in Figure 2, our lightweight Learnable Importance Scorer is designed to capture each token's relative importance in the global visual context. It uses two linear layers to project the input token features $V$ into a Query ($Q$) and a Key ($K$):

$$Q = VW_q, K = VW_k, \tag{1}$$

where $W_k, W_k \in \mathbb{R}^{D \times d}$ are learnable projection matrices, and $d$ is the hidden dimension.

We then compute the matrix product of $Q$ and $K$ to obtain a simplified self-attention score matrix $A \in \mathbb{R}^{N \times N}$. The final importance score $s_i$ for each token $i$ is the average of its interaction scores with all other tokens:

$$A = \frac{QK^T}{\sqrt{d}}, s_i = \frac{1}{N}\sum_{j=1}^{N} A_{ij}. \tag{2}$$

This design enables the scorer to perceive global context while remaining computationally efficient. To ensure a smooth integration of the LIS module with the frozen pre-trained model and to maintain stability during the initial training phase, we adopt a near-zero initialization strategy for its weights.

### 3.3 DIFFERENTIABLE TOP-K SELECTION MECHANISM

The standard Top-K operator is discrete and non-differentiable, which prevents end-to-end gradient flow to the LIS module. Inspired by previous work(Xie et al., 2020; Petersen et al., 2022b; Sander et al., 2023; Struski et al., 2025; Ahle, 2023; Zhu et al., 2025), we adopt a continuous relaxation of Top-K to produce a soft mask during training, $M_{soft} = \text{DiffTopK}(s) \in (0,1)^{B \times N}$, where $\text{DiffTopK}$ denotes the differentiable Top-K operator. The procedure is summarized in Algorithm1.

#### 3.3.1 FORWARD PASS

Given a score vector $s \in \mathbb{R}^{B \times N}$ and the desired number of retained tokens $k = N \times b$, we search for a scalar threshold $t$ via bisection such that:

$$M = \text{DiffTopK}(s) = \sigma(s + t), \sum_{i=1}^{N} M_i \approx k, \tag{3}$$

where $\sigma(\cdot)$ is the sigmoid function. The resulting soft mask $M_{soft}$ can be interpreted as per-token selection probabilities.Unlike Gumbel-Softmax, which introduces stochastic perturbations during training and lacks strict monotonicity, the operator $s \rightarrow M = \text{DiffTopK}(s)$ inherits the monotonicity of the sigmoid. Specifically, for any indices $i, j \in \{1, 2, ..., N\}$,

$$[\text{DiffTopK}(s)]_i > [\text{DiffTopK}(s)]_j \Leftrightarrow s_i > s_j, \tag{4}$$

so high-score tokens are promoted by the soft mask and low-score tokens are suppressed.

#### 3.3.2 BACKWARD PASS VIA IMPLICIT DIFFERENTIATION

We differentiate $M = \sigma(s + t)$ under the implicit constraint $\sum_{i=1}^{N} \sigma(s_i + t) = k$. Let

$$v = \sigma'(s + t), v_i = M_i(1 - M_i). \tag{5}$$

From the constraint, we obtain

$$\frac{\partial t}{\partial s_j} = -\frac{v_j}{\sum_i v_i}, \tag{6}$$

which yields the Jacobian of the output with respect to the input:

$$\frac{\partial M}{\partial s} = diag(v) - \frac{vv^T}{\sum_i v_i}. \tag{7}$$

Therefore, for any upstream gradient $g = \frac{\partial L}{\partial s}$,

$$\frac{\partial L}{\partial s} = v \odot g - \frac{v^T g}{\sum_i v_i} v, \tag{8}$$

enabling end-to-end gradient propagation. The bisection is used only to solve for the threshold $t$ in the forward pass. Gradients are provided by implicit differentiation rather than backpropagating through the search.

#### 3.3.3 INFERENCE

This continuous relaxation is used only during training for gradient propagation. At inference time, we apply the standard Top-K operator directly to the score vector to obtain a hard binary mask that selects the top $k$ tokens.

### 3.4 Training Objective with Curriculum Annealing Strategy

To effectively train the LIS and bridge the discrepancy between soft selection during training and hard selection during inference, we design a composite loss function and introduce a curriculum learning-based weight scheduling strategy. The total loss function $L_{total}$ is defined as:

$$L_{total} = L_{CE} + \lambda_t L_{constraint}, \tag{9}$$

where $L_{CE}$ is the cross-entropy loss for the downstream task, $L_{constraint}$ is a constraint loss to regularize the selection process, and $\lambda_t$ is a dynamic weighting coefficient that is adjusted with the training step $t$.

The objective of $L_{constraint}$ is to guide the soft mask $M_{soft}$ used during training, to approximate the ideal hard mask $M_{hard}$ used at inference. We employ a binary cross-entropy loss to measure the discrepancy between them:

$$M_{hard} = TopK(s), \tag{10}$$

$$L_{constraint} = BCE(M_{soft}, M_{hard}). \tag{11}$$

The $L_{constraint}$ incentivizes the output values of the soft mask $M_{soft}$ to polarize towards either 0 or 1, thereby enabling the score distribution learned by the LIS to be directly and effectively applied for hard selection during inference.

The weighting coefficient $\lambda_t$ is adjusted using a Curriculum Annealing Strategy. Early in training, $\lambda_t$ is set to a small initial value $\lambda_{start}$, allowing the model to primarily focus on learning the downstream task itself. As training progresses, $\lambda_t$ linearly increases to a predefined maximum value $\lambda_{end}$, according to the schedule:

$$\lambda_t = \lambda_{start} + (\lambda_{end} - \lambda_{start}) \times min(\frac{t_{current}}{t_{total}}, 1.0), \tag{12}$$

where $t_{current}$ is the current training step, and $t_{total}$ is the total number of training steps.

This Curriculum Annealing Strategy ensures that the model first masters the fundamentals of the task before gradually strengthening the regularization constraint on the token selection process, thereby achieving a more stable and effective joint optimization.

## 4 Experiments

To validate the effectiveness of VisionSelector, we conduct a series of experiments. This section details our experimental setup, implementation specifics, and the evaluation benchmarks used.

### 4.1 Experimental Setting

**Models, Training Data and Hardware.** All our experiments are based on the Qwen2.5-VL-7B(Bai et al., 2025), a powerful open-source MLLM that serves as a high-performance baseline for our study. We conduct supervised training by integrating our proposed LIS module into this model. For training, we employ a mixed-dataset strategy. Our training data is a composite of several datasets from Cambrian-737K (Tong et al., 2024), totaling approximately 144K samples. It primarily includes ChartQA (Masry et al., 2022), OCRVQA (Mishra et al., 2019), and a 10% random sample of the COCO (Lin et al., 2014) dataset. This composition exposes the model to diverse visual scenarios, including chart understanding, document OCR, and natural images, which enhances its generalization capability. All experiments are conducted on 8 NVIDIA A800 GPUs (80GB). We utilized a distributed data-parallel strategy and leveraged DeepSpeed ZeRO Stage 3 (Rasley et al., 2020) to optimize the training process for efficient management of memory and computational resources.

**Comparison Methods, Evaluation Framework, Datasets, and Tasks.** We select the attention-based methods FastV (Chen et al., 2024), PruMerge+ (Shang et al., 2024) and VisionZip (Yang et al., 2025b), as well as the non-attention-based methods DART (Wen et al., 2025) and DivPrune (Alvar et al., 2025) as our baselines. See the Appendix A.2 for more details. To ensure fairness and reproducibility, we conduct a comprehensive evaluation of our method and the comparison methods under the same LMMs-Eval (Zhang et al., 2024a) framework. Our evaluation covers both image and

video modalities, utilizing 9 image-language datasets and 4 video-language datasets. **For image-language understanding**, we focus on assessing performance in information-dense visual tasks. We select a suite of OCR and text-centric VQA datasets, including TextVQA (Singh et al., 2019), DocVQA (Mathew et al., 2021), OCRBench (Liu et al., 2024a), and ChartQA (Masry et al., 2022), to test the model's ability to recognize and comprehend embedded text. To measure complex reasoning about object relationships, attributes, and spatial layouts, we employ the AI2D (Kembhavi et al., 2016) and ScienceQA (Lu et al., 2022) datasets. Furthermore, we test cross-domain knowledge and expert-level reasoning using two challenging comprehensive benchmarks: MME (Fu et al., 2024) and MMMU (Yue et al., 2024). Finally, we used the POPE (Li et al., 2023c) benchmark to quantify the model's hallucination levels and ensure content faithfulness. **For video-language understanding**, we utilized MVBench (Li et al., 2024), SEEDBench (Li et al., 2023a), VideoMME (Fu et al., 2025), and NeXT-QA (Xiao et al., 2021). These benchmarks collectively form a comprehensive evaluation suite designed to measure performance on advanced tasks such as temporal event understanding, dynamic relationship reasoning, and causal question-answering over video content.

### 4.2 IMPLEMENTATION DETAILS

**Model Configuration and Training Strategy.** We seamlessly integrate the LIS module following the modality interface of the baseline model. During training, we adopt a parameter-efficient approach, exclusively updating the parameters of the LIS module while keeping all pre-trained weights frozen. This strategy not only preserves the powerful pre-trained knowledge of the baseline model and is suitable for scenarios with proprietary training sets, but also significantly reduces the computational overhead of training.

**Hyperparameter Settings.** We train the model for 1 epoch on the mixed dataset. The projection dimension $d$ for $W_q$ and $W_k$ is set to 1792. We use the AdamW optimizer with a cosine annealing learning rate scheduler, setting the initial learning rate to $5e - 5$ with a linear warm-up for the first 0.03 epochs. The per-device batch size is 16, and we use 4 steps of gradient accumulation, resulting in an effective global batch size of 256. The retention budget for visual tokens is set to 20%. For the constraint loss weight, $\lambda_t$, we adopt a Curriculum Annealing Strategy, linearly increasing it from an initial value of 0.1 to a final value of 2.0 over the course of training.

### 4.3 IMAGE-LANGUAGE UNDERSTANDING EVALUATION

To evaluate VisionSelector, we compare a model trained with a 20% budget against multiple baselines across varying budgets, and the results in Table 1 validate the effectiveness of our method while highlighting the advantage of an end-to-end learning paradigm over fixed, heuristic rules.

The experimental data clearly indicate that across all compression ratios, the overall performance of VisionSelector significantly surpasses that of all baseline methods. At a 20% budget, VisionSelector maintains 94.83% of the baseline model's average performance, outperforming the next-best method by over 7 percentage points. Under an extreme compression budget of 10%, this performance gap widens to more than 12 percentage points. Particularly noteworthy is the behavior when the budget tightens from 20% to 10%: attention-based baselines show a sharp performance drop, with VisionZip decreasing by approximately 14 percentage points, whereas VisionSelector's performance declines more gracefully by about 6 percentage points. We hypothesize that the severe performance drop in baseline methods is attributable to their reliance on pre-trained attention maps. When the budget is extremely limited, inherent biases such as attention sink may force these models to retain tokens that are positionally early but semantically irrelevant, leading to a performance collapse. In contrast, the learnable mechanism of VisionSelector demonstrates superior robustness.

A surprising finding is that in certain scenarios, VisionSelector not only enhances efficiency but can also improve performance beyond the 100% token baseline by filtering out noisy information. With a 30% budget, VisionSelector achieves 100.07% relative performance on the MME benchmark, demonstrating lossless and even gainful compression. This phenomenon indicates that our learned importance scores enable VisionSelector to effectively prune task-irrelevant and potentially distracting visual noise. Consequently, the model can better focus on critical information, leading to improved reasoning accuracy.

Table 1: Comparison results with different methods on Image-Language benchmarks. Note that our method, trained with a fixed 20% retention budget, exhibits adaptability to varying compression budgets during inference. Evaluation is conducted with LMMs-Eval (Zhang et al., 2024a).

| Method | DocVQA Anls | ChartQA Relaxed | TextVQA EM | OCRBench Acc | ScienceQA EM | AI2D EM | MMMU Acc | MME Score | POPE F1 | Avg |
|---|---|---|---|---|---|---|---|---|---|---|
| *Dynamic Resolution(MinPix=256×28×28,MaxPix=2048×28×28),Upper Bound (100%)* | | | | | | | | | | |
| Avg. Visual Tokens | 1951.61 | 596.06 | 976.58 | 652.82 | 323.05 | 510.19 | 601.15 | 867.67 | 359.55 | |
| Qwen-2.5-VL-7B | *94.33* | 83.40 | 82.84 | 838 | 87.26 | 93.59 | 50.78 | 2342.15 | 86.19 | 100% |
| *Retain 30% Tokens (70% Compression Ratio)* | | | | | | | | | | |
| FastV (ECCV2024) | 84.01 89.06% | 67.64 81.10% | 80.22 96.84% | 687 81.98% | 83.06 95.22% | 86.92 92.87% | 49.44 97.36% | 2263.58 96.65% | 80.47 93.36% | 91.61% |
| PruMerge+ (ICCV2025) | 73.95 78.39% | 62.24 74.63% | 73.71 88.98% | 648 77.33% | 85.22 97.66% | 82.77 88.44% | 47.67 93.88% | 2239.64 95.62% | 83.69 97.10% | 88.00% |
| VisionZip (CVPR2025) | 86.11 91.29% | 72.28 86.67% | 77.30 93.31% | 711 84.84% | 86.61 99.26% | 87.86 93.88% | 49.44 97.36% | 2276.04 97.18% | 84.73 98.31% | 93.57% |
| DART (EMNLP2025) | 62.60 66.36% | 56.88 68.20% | 74.45 89.87% | 629 75.06% | 84.33 96.64% | 75.94 81.14% | 47.89 94.31% | 2218.83 94.73% | 83.43 96.80% | 84.79% |
| DivPrune (CVPR2025) | 82.51 87.47% | 67.52 80.96% | 78.52 94.79% | 720 85.92% | 86.01 98.57% | 88.28 94.33% | 48.33 95.18% | 2224.06 94.96% | 84.68 98.25% | 92.27% |
| **VisionSelector** | **92.89** 98.47% | **72.96** 87.48% | **81.45** 98.32% | **809** 96.54% | 85.77 98.29% | **92.00** 98.30% | 50.11 98.68% | 2343.77 100.07% | 85.05 98.68% | **97.20%** |
| *Retain 20% Tokens (80% Compression Ratio)* | | | | | | | | | | |
| FastV (ECCV2024) | 75.99 80.56% | 60.48 72.52% | 78.01 94.17% | 597 71.24% | 82.35 94.83% | 82.35 87.99% | 49.00 96.49% | 2152.74 91.91% | 76.12 88.32% | 86.45% |
| PruMerge+ (ICCV2025) | 61.09 64.76% | 52.56 63.02% | 68.72 82.96% | 562 67.06% | 83.79 96.02% | 77.62 82.94% | 46.11 90.80% | 2219.3 94.75% | 81.74 94.84% | 81.91% |
| VisionZip (CVPR2025) | 74.75 79.24% | 62.04 74.39% | 72.03 86.95% | 591 70.53% | 84.68 97.04% | 82.32 87.96% | 47.00 92.56% | 2168.86 92.60% | 83.23 96.57% | 86.43% |
| DART (EMNLP2025) | 50.03 53.04% | 47.16 56.55% | 67.53 81.52% | 537 64.08% | 83.34 95.51% | 71.04 75.91% | 47.00 92.56% | 2138.61 91.31% | 80.16 93.00% | 78.16% |
| DivPrune (CVPR2025) | 73.81 78.25% | 57.88 69.40% | 73.86 89.16% | 648 77.33% | 84.33 96.64% | 82.29 87.93% | 46.33 91.24% | 2198.82 93.88% | 83.55 96.94% | 86.75% |
| **VisionSelector** | **89.91** 95.31% | **68.84** 82.54% | **80.05** 96.63% | **770** 91.89% | 85.67 98.18% | **90.15** 96.32% | 49.22 96.93% | 2293.54 97.92% | 84.27 97.77% | **94.83%** |
| *Retain 10% Tokens (90% Compression Ratio)* | | | | | | | | | | |
| FastV (ECCV2024) | 58.64 62.16% | 44.64 53.53% | 70.83 85.50% | 440 52.51% | 81.95 93.91% | 75.74 80.93% | 45.78 90.15% | 1940.91 82.87% | 65.99 76.56% | 75.35% |
| PruMerge+ (ICCV2025) | 42.08 44.61% | 41.56 49.83% | 56.87 68.65% | 417 49.76% | 81.56 93.47% | 71.08 75.95% | 45.22 89.05% | 1948.58 83.20% | 76.52 88.78% | 71.48% |
| VisionZip (CVPR2025) | 48.29 51.19% | 42.84 51.37% | 55.94 67.53% | 404 48.21% | 82.60 94.66% | 73.09 78.10% | 45.44 89.48% | 1944.04 83.00% | 78.46 91.03% | 72.73% |
| DART (EMNLP2025) | 33.13 35.12% | 34.00 40.77% | 53.97 65.15% | 415 49.52% | 81.85 93.80% | 67.10 71.70% | 46.44 91.45% | 1980.7 84.57% | 71.91 83.43% | 68.39% |
| DivPrune (CVPR2025) | 54.04 57.29% | 39.80 47.72% | 64.65 78.04% | 477 56.92% | 82.15 94.14% | 72.80 77.79% | 46.00 90.59% | 2015.66 86.06% | 79.27 91.97% | 75.61% |
| **VisionSelector** | **77.25** 81.89% | **62.28** 74.68% | **75.37** 90.98% | **646** 77.09% | 83.54 95.74% | **84.72** 90.52% | 47.44 93.42% | 2175.75 92.90% | 79.73 92.50% | **87.75%** |

Table 2: Performance and efficiency comparisons. Task performance is evaluated across various video-language benchmarks, while efficiency metrics are benchmarked on MVBench.

| Method | MVBench Acc | SEEDBench Acc | VideoMME Score | NextQA WUPS | Performance (%) | Max GPU mem (GB) | Prefill Time (ms) | E2E Latency (ms) |
|---|---|---|---|---|---|---|---|---|
| Qwen2.5-VL-7B | 68.10 | 60.48 | 60.67 | 27.58 | 100% | 25.97 | 1413.34 | 1605.31 |
| FastV (ECCV2024) | 65.75 | *oom* | 59.15 | 27.00 | 97.31% | 24.63 | 851.76 | 1021.83 |
| DART (EMNLP2025) | 65.80 | **61.00** | 57.74 | 26.84 | 97.49% | 18.93 | 832.58 | 996.55 |
| Divprune (CVPR2025) | 65.85 | 59.79 | 57.78 | 27.00 | 97.17% | **17.55** | 1184.00 | 1345.24 |
| Ours | **66.55** | 59.82 | **59.22** | **27.10** | **98.13%** | 17.57 | **760.82** | **924.57** |

## 4.4 VIDEO-LANGUAGE UNDERSTANDING EVALUATION

To further validate the generalization capability of VisionSelector, we evaluate it on video–language understanding tasks. Specifically, we take the model trained exclusively on image data with a 20% budget and directly apply it to four representative video benchmarks: MVBench, SEED-Bench, VideoMME, and NeXT-QA.

In selecting baselines, we note that VisionZip and PruMerge+ cause out-of-memory (OOM) errors when integrated with the Qwen2.5-VL architecture, owing to their operational placement. Qwen2.5-VL utilizes a PatchMerger module for initial token compression after the visual encoder. Since both VisionZip and PruMerge+ compute attention at the direct output of the visual encoder, they incur prohibitive computational and memory costs. Consequently, we select FastV, DART, and DivPrune as baselines for this evaluation.

The results in Table 2 show that VisionSelector outperforms FastV, DART and DivPrune across most video datasets. On MVBench, VisionSelector achieves 66.55% accuracy, significantly higher than the baselines. Overall, VisionSelector attains a 98.13% performance retention rate on these video

Table 3: Ablation study on the impact of training data composition and curriculum annealing strategy ($\lambda_t$) on model performance.

| Config | Dataset | | | $\lambda_t$ | BatchSize | LearningRate | Dim | DocVQA | OCRBench | MMMU | MME | POPE | Avg |
|---|---|---|---|---|---|---|---|---|---|---|---|---|---|
| | OCRVQA | ChartQA | 10%COCO | | | | | Anls | Acc | Acc | Score | F1 | |
| 1 | ✓ | | | 0.1~3 | 8 | 1e-4 | 1024 | 86.85 | 700 | 48.78 | 2238.43 | 81.58 | 92.38% |
| 2 | ✓ | ✓ | | 0.1~3 | 8 | 1e-4 | 1024 | 87.59 | 701 | 48.78 | 2270.09 | 81.84 | 92.89% |
| 3 | ✓ | ✓ | ✓ | 0.1~3 | 8 | 1e-4 | 1024 | 89.34 | 763 | 48.89 | 2282.37 | 83.93 | 95.37% |
| 4 | ✓ | ✓ | ✓ | 0 | 8 | 1e-4 | 1024 | 88.88 | 760 | 47.67 | 2284.06 | 83.43 | 94.62% |
| 5 | ✓ | ✓ | ✓ | 3 | 8 | 1e-4 | 1024 | 83.69 | 613 | 47.22 | 2172.38 | 83.70 | 88.94% |
| 6 | ✓ | ✓ | ✓ | 0.1~2 | 8 | 1e-4 | 1024 | 89.54 | **779** | 49.00 | 2272.71 | 83.90 | 95.75% |
| VisionSelector | ✓ | ✓ | ✓ | 0.1~2 | 16 | 5e-5 | 1792 | **89.91** | 770 | **49.22** | **2293.54** | **84.27** | **95.96%** |

tasks. This demonstrates that importance criteria learned through an end-to-end process generalize more effectively to temporal data and enable the precise identification of key video tokens compared to fixed heuristic rules.

### 4.5 EFFICIENCY ANALYSIS

This section quantifies the computational efficiency advantages of our method. We analyze efficiency on video tasks across three key metrics: max memory usage, prefill time, and end-to-end (E2E) latency.

As detailed in Table 2, we analyze this using the MVBench dataset, which has a high average token count of 6,828. VisionSelector's prefill time is only 760.82 ms and its end-to-end latency is 924.57 ms, achieving $1.86\times$ and $1.74\times$ speedups over the baseline model, respectively, and is significantly faster than all comparison methods. Our method matches DivPrune in memory efficiency by lowering peak memory usage to $17.57$ GB, a $32.3\%$ reduction from the baseline model. This implies that under the same hardware constraints, our method can process longer visual contexts than competing methods, which is crucial for advancing the application of MLLMs in real-world scenarios like long-video understanding.

### 4.6 ABLATION STUDY

We conduct a series of comprehensive ablation studies to validate the effectiveness of VisionSelector's key components and to determine the optimal hyperparameter settings. All experiments are performed on the Qwen2.5-VL-7B model, and we evaluate the results across several benchmarks spanning multiple dimensions: document understanding (DocVQA, OCRBench), scientific reasoning (ScienceQA, MMMU), and hallucination evaluation (MME, POPE). The detailed configurations and results are presented in Table 3.

**Impact on training data composition.** As detailed in configurations 1, 2, and 3, we progressively enrich the training data. The results reveal a clear trend: a more diverse training set leads to significant performance gains. Notably, with the inclusion of the general-purpose COCO dataset (Config 3 vs. Config 2), the model's performance on OCR-related tasks improves substantially: the ANLS score on DocVQA increases from 87.59 to 89.34, and the OCRBench score jumps from 701 to 763. This indicates that exposure to diverse natural images enhances the model's general visual representation capabilities, which in turn benefits its performance on domain-specific tasks. Furthermore, the increase in the F1-score on the POPE dataset suggests that diverse training data also helps to suppress model hallucinations.

**Impact of the Curriculum Annealing Strategy for $\lambda_t$.** To validate the importance of the Curriculum Annealing Strategy, we compare the optimal annealing setting in Config 6 against alternative weighting schemes. We first observe that eliminating the constraint loss in Config 4 results in a suboptimal average score of 94.62% compared to the 95.75% achieved by Config 6. This outcome demonstrates that explicit alignment with the hard mask effectively enhances optimization stability and ensures consistent gains. Conversely, applying a constant high weight in Config 5 causes the average score to drop sharply to $88.94\%$. This finding confirms that an excessively high initial constraint loss forces the model to prematurely focus on polarizing token scores rather than on learning the downstream task itself.

### 4.7 VISUALIZATION OF TOKEN SELECTION RESULTS

As illustrated in Figure 3, heuristic-based pruning methods exhibit clear limitations. VisionZip is prone to issues like attention sink and dispersion, which obscure salient tokens. DivPrune tends to

drop semantically rich tokens, such as the logo. Both methods incorrectly preserve low-information background tokens, highlighting their dependence on model-specific feature distributions. In contrast, VisionSelector pinpoints and retains the critical tokens containing the phone number to answer correctly, demonstrating the superiority of its learned selection mechanism.

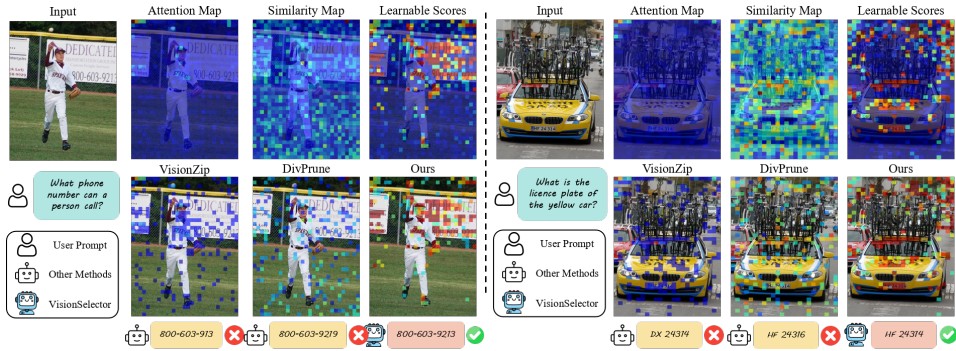

Figure 3: Qualitative results of VisionSelector on TextVQA. The top row compares three methods for scoring token importance, where **warmer colors (red) correspond to higher importance scores**. The bottom row illustrates the tokens selected by each method and the corresponding model predictions at a 20% compression budget.

## 5  LIMITATIONS

While our study demonstrates the significant potential of token compression for MLLM inference acceleration, it shares an inherent limitation with current selection-based paradigms: it operates on a 'hard selection' (i.e., keep-or-discard) principle. This mechanism, while effective, inevitably results in the complete loss of information from discarded tokens. Furthermore, failure cases appear under extreme compression rates where fixed token budgets occasionally fail to cover all necessary semantics in images containing multiple spatially scattered objects. This limitation points toward a key direction for future research: exploring new token compression paradigms to achieve lossless performance.

## 6  CONCLUSION

In this study, we present VisionSelector, a learnable token pruning method based on a differentiable Top-K mechanism. During training, the model assigns importance scores to visual tokens via a learnable scorer, applies a differentiable Top-K to produce a soft mask, and uses a hard-mask constraint with curriculum annealing to bridge the train–inference gap. At inference, we revert to a standard and efficient Top-K selection. Driven by downstream objectives, the model autonomously discovers critical visual tokens. Comprehensive evaluations on various image and video benchmarks demonstrate that VisionSelector sets a new state-of-the-art. It provides a superior balance of inference speed, memory footprint, and model accuracy across various compression budgets.

### REPRODUCIBILITY STATEMENT

Data preparation appears in Section 4.1, and to ensure reproducible sampling on the COCO dataset, we fix the random seed at 42 during training. Implementation details appear in Section 4.2. Code and models will be publicly available.

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

# A  APPENDIX

## A.1  COMPARISON WITH LEARNABLE ALTERNATIVES ON QWEN2.5-VL-7B

### A.1.1  COMPARISON WITH DYNAMIC-LLAVA

To ensure a fair comparison and validate the effectiveness of our approach, we re-implemented the trainable image predictor module from Dynamic-LLaVA (Huang et al.) (referred to as Dynamic) on Qwen2.5-VL-7B. During training, only the parameters of the image predictor are updated, while all other components remain frozen. The training configuration strictly follows the settings described in the original paper: the temperature of the Gumbel Softmax is annealed from $\tau_{start} = 1.0$ to $\tau_{end} = 0.1$, the learning rate of the image predictor is set to $2e-4$, and the batch size is 8. The model is trained for 1 epoch on the same datasets used by our method, including OCRVQA, TextVQA, and 10% randomly sampled COCO images.

The results are summarized in Table 1 and Table 4. Compared with training-free methods, Dynamic demonstrates strong capability on text-oriented benchmarks (e.g., DocVQA, ChartQA, TextVQA, and OCRBench), though its generalization to broader visual reasoning datasets such as MMMU and POPE is somewhat reduced. In contrast, VisionSelector consistently surpasses Dynamic across all benchmarks and also outperforms training-free methods in most cases. Remarkably, it maintains strong robustness and generalization even under severe compression (retaining only 20% or 10% of tokens), demonstrating its superior capability in adaptive visual token selection.

### A.1.2  COMPARISON WITH STE-STYLE PRUNING

The Straight-Through Estimator (STE) (Bengio et al., 2013) is a widely adopted technique for estimating gradients through non-differentiable discrete operations. To verify the effectiveness of our proposed operator, we implement an STE-style pruning baseline inspired by LightVLA (Jiang et al., 2025), substituting our differentiable Top-K mechanism with their selection strategy while maintaining identical experimental settings.

Specifically, after obtaining the importance score map $A \in \mathbb{R}^{N \times N}$ from the LIS, we convert it into an indicator matrix $I \in \mathbb{R}^{N \times N}$. The process involves generating both a soft differentiable score and a hard discrete mask:

$$A_{soft} = \mathrm{softmax}_j(A), \tag{13}$$

$$A_{hard} = \mathrm{one} - \mathrm{hot}(\mathrm{argmax}_j(A)), \tag{14}$$

$$I = A_{hard} + A_{soft} - A_{soft}^{SG}, \tag{15}$$

where $SG$ denotes the stop-gradient operation. In this formulation, $A_{hard}$ represents the binary selection mask used during the forward pass, while gradients are propagated through $A_{soft}$ during backpropagation. The pruned visual tokens are then computed as $V_{pruned} = IV^T$, retaining their original positional embeddings. Throughout this process, only the LIS module is trainable, with all other components frozen.

The suboptimal results of STE-style pruning in Table 4 likely stem from the absence of global sorting and the inherent gradient mismatch issue (Bengio et al., 2013; Jang et al., 2017). Conversely, our DiffTopK approach ensures theoretically guaranteed gradient stability and monotonicity, as demonstrated in DFTopK (Zhu et al., 2025) and A.9.3, enabling more effective end-to-end optimization.

### A.1.3  COMPARISON WITH SOFTSORT

To investigate the versatility of the VisionSelector framework and the specific contribution of the Differentiable Top-K operator, we conduct an ablation study by replacing the Differentiable Top-K module with the established SoftSort operator (Prillo & Eisenschlos, 2020). We maintain identical network architectures and hyperparameters to ensure that any performance variations result

solely from the ranking approximation mechanism. Table 4 presents the comparative results on the Qwen2.5-VL-7B backbone. The SoftSort-based variant consistently outperforms heuristic-based methods and the learnable Dynamic approach across all tested token retention rates. Specifically, it achieves substantial average performance gains ranging from 3.58% to 7.41% over Dynamic. Furthermore, the SoftSort variant yields results comparable to our proposed DiffTop-K implementation and even exhibits marginal superiority on ScienceQA. However, the standard VisionSelector with DiffTop-K generally offers better stability and generalization. This advantage arises because the DiffTop-K operator effectively mitigates gradient conflicts and resolves the sparsity gap between training and inference phases. These findings demonstrate that the VisionSelector framework is compatible with various differentiable operators and does not rely exclusively on a single sorting mechanism. Even with SoftSort, our framework provides stable performance improvements and surpasses baselines such as sparse selection or Gumbel-Softmax gating.

Table 4: Comparison results of our method and learnable alternatives on image-language understanding datasets under Qwen2.5-VL-7B.

| Method | DocVQA Anls | ChartQA Relaxed | TextVQA EM | OCRBench Acc | ScienceQA EM | AI2D EM | MMMU Acc | MME Score | POPE F1 | Avg |
|---|---|---|---|---|---|---|---|---|---|---|
| *Dynamic Resolution(MinPix=256×28×28,MaxPix=2048×28×28),Upper Bound (100%)* | | | | | | | | | | |
| Avg. Visual Tokens | 1951.61 | 596.06 | 976.58 | 652.82 | 323.05 | 510.19 | 601.15 | 867.67 | 359.55 | |
| Qwen-2.5-VL-7B | 94.33 | 83.40 | 82.84 | 838 | 87.26 | 93.56 | 50.78 | 2342.15 | 86.19 | 100% |
| *Retain 30% Tokens (70% Compression Ratio)* | | | | | | | | | | |
| Dynamic (ICLR2025) | 86.32 91.51% | 68.88 82.59% | 73.56 88.80% | 750 89.50% | 78.78 90.28% | 83.29 88.99% | 41.78 82.28% | 2180.42 93.09% | 80.87 93.83% | 88.99% |
| STE-style Pruning (Arxiv2025) | 71.09 75.36% | 49.72 59.62% | 71.74 86.60% | 559 66.71% | 83.79 96.02% | 80.54 86.06% | 46.89 92.34% | 2204.46 94.12% | 84.28 97.78% | 83.85% |
| SoftSort (ICML2020) | 92.00 97.53% | 72.84 87.34% | 80.67 97.38% | 783 93.44% | 86.07 98.64% | 91.45 97.71% | 48.78 96.06% | 2311.28 98.68% | 84.05 97.52% | 96.03% |
| **VisionSelector** | **92.89** 98.47% | **72.96** 87.48% | **81.45** 98.32% | **809** 96.54% | **85.77** 98.29% | **92.00** 98.30% | **50.11** 98.68% | **2343.77** 100.07% | **85.05** 98.68% | **97.20%** |
| *Retain 20% Tokens (80% Compression Ratio)* | | | | | | | | | | |
| Dynamic (ICLR2025) | 79.21 83.97% | 65.92 79.04% | 71.73 86.59% | 674 80.43% | 77.00 88.24% | 81.87 87.48% | 42.56 83.81% | 2117.37 90.40% | 77.22 89.59% | 85.51% |
| STE-style Pruning (Arxiv2025) | 57.48 60.94% | 39.48 47.34% | 59.93 72.34% | 428 51.07% | 82.99 95.11% | 75.16 80.31% | 45.56 89.72% | 1985.39 84.77% | 81.71 94.80% | 75.16% |
| SoftSort (ICML2020) | 86.90 92.12% | 68.20 81.77% | 77.76 93.87% | 731 87.23% | 86.22 98.81% | 89.15 95.26% | 48.11 94.74% | 2259.01 96.45% | 82.77 96.03% | 92.92% |
| **VisionSelector** | **89.91** 95.31% | **68.84** 82.54% | **80.05** 96.63% | **770** 91.89% | **85.67** 98.18% | **90.15** 96.32% | **49.22** 96.93% | **2293.54** 97.92% | **84.27** 97.77% | **94.83%** |
| *Retain 10% Tokens (90% Compression Ratio)* | | | | | | | | | | |
| Dynamic (ICLR2025) | 61.17 64.85% | 59.96 71.89% | 68.25 82.39% | 557 66.47% | 75.71 86.76% | 76.10 81.31% | 43.11 84.90% | 1977.58 84.43% | 70.80 82.14% | 78.35% |
| STE-style Pruning (Arxiv2025) | 36.32 38.50% | 22.52 27.00% | 37.22 44.93% | 244 29.12% | 78.38 89.82% | 68.91 73.63% | 44.56 87.75% | 1744.56 74.49% | 75.55 87.66% | 61.43% |
| SoftSort (ICML2020) | 69.61 73.79% | 58.24 69.83% | 65.62 79.21% | 546 65.16% | 82.40 94.43% | 81.09 86.64% | 46.56 91.69% | 2014.89 86.03% | 78.09 90.60% | 81.93% |
| **VisionSelector** | **77.25** 81.89% | **62.28** 74.68% | **75.37** 90.98% | **646** 77.09% | **83.54** 95.74% | **84.72** 90.52% | **47.44** 93.42% | **2175.75** 92.90% | **79.73** 92.50% | **87.75%** |

## A.2 DETAILS ABOUT COMPARISON METHOD

We select five representative visual token compression methods for comparison, covering several mainstream technical approaches. FastV (Chen et al., 2024) is an attention-based pruning method that operates after the second layer of the MLLM. It selects visual tokens based on the attention scores they receive from text tokens. PruMerge+ (Shang et al., 2024) also leverages attention mechanisms but at the visual encoder stage. It identifies key tokens via attention sparsity and then merges the remaining tokens using K-Nearest Neighbors (KNN) clustering. VisionZip (Yang et al., 2025b) is a text-agnostic compression method. It selects "dominant tokens" based on the attention map from the final visual encoder layer and merges the remaining "contextual tokens" according to semantic similarity. DART (Wen et al., 2025) identifies groups of near-duplicate tokens by computing the cosine similarity between all visual tokens. It then retains only one representative token from each group, achieving efficient compression while preserving key information. DivPrune (Alvar et al., 2025) models the token pruning task as a Max-Min Diversity Problem, aiming to maximize the informational diversity of the preserved subset of tokens.

## A.3 ADDITIONAL EXPERIMENTS ON ABLATION STUDY

To investigate the impact of hyperparameters (batch size, learning rate) and the score projector dimension $d$, we provide a detailed ablation study in Table 5.

**Impact of Hyperparameters.** As we increase the batch size from 8 (Config 5) to 16 (Config 6), we observe a modest but consistent performance improvement. However, a further increase to 32 (Config 7) leads to a slight performance decline. This indicates that a batch size of 16 is optimal for our setup, likely due to more stable gradient estimation. Subsequently, we adjust the learning rate. Lowering the learning rate from $1e-4$ to $5e-5$ (Config 7 vs. Config 8) yields further performance gains, especially on complex reasoning benchmarks such as ScienceQA and MMMU. This suggests that a smaller learning rate facilitates better model convergence for our complex, joint optimization objective.

**Impact of Scorer Projection Dimension $d$.** We increase $d$ from 1024 (Config 8) to 1792 (Config 11), which corresponds to half of the model's hidden dimension. This adjustment leads the model to achieve its best performance across all configurations, reaching a peak average score of 96.33%. The performance gain is particularly pronounced on the POPE benchmark, where the F1-score peaks at 84.27. This indicates that a larger-capacity scorer can more finely model complex inter-token relationships, leading to more accurate identification of critical visual information and more effective suppression of object-level hallucinations.

**Impact of Initialization.** We replace the near zero initialization with random initialization (Config 9) and Kaiming initialization (Config 10). Across the six evaluation benchmarks their overall performance remains very close to that with near zero initialization (Config 11) and all configurations converge stably. This observation indicates that our scorer is not sensitive to the specific choice of parameter initialization.

Table 5: Ablation study on the impact of training data composition, curriculum annealing strategy ($\lambda_t$), and hyperparameters on model performance.

| Config | Dataset | | | $\lambda_t$ | BatchSize | LearningRate | Dim | Init | DocVQA Anls | OCRBench Acc | ScienceQA EM | MMMU Acc | MME Score | POPE F1 | Avg |
|---|---|---|---|---|---|---|---|---|---|---|---|---|---|---|---|
| | OCRVQA | ChartQA | COCO | | | | | | | | | | | | |
| 1 | ✓ | | | 0.1~3 | 8 | 1e-4 | 1024 | Near Zero | 86.85 | 700 | 85.47 | 48.78 | 2238.43 | 81.58 | 93.31% |
| 2 | ✓ | ✓ | | 0.1~3 | 8 | 1e-4 | 1024 | Near Zero | 87.59 | 701 | 84.78 | 48.78 | 2270.09 | 81.84 | 93.60% |
| 3 | ✓ | ✓ | ✓ | 0.1~3 | 8 | 1e-4 | 1024 | Near Zero | 89.34 | 763 | 85.52 | 48.89 | 2282.37 | 83.93 | 95.81% |
| 4 | ✓ | ✓ | ✓ | 3 | 8 | 1e-4 | 1024 | Near Zero | 83.69 | 613 | 84.48 | 47.22 | 2172.38 | 83.70 | 90.26% |
| 5 | ✓ | ✓ | ✓ | 0.1~2 | 8 | 1e-4 | 1024 | Near Zero | 89.54 | 779 | 84.68 | 49.00 | 2272.71 | 83.90 | 95.97% |
| 6 | ✓ | ✓ | ✓ | 0.1~2 | 16 | 1e-4 | 1024 | Near Zero | 89.98 | 772 | 84.88 | 49.22 | 2279.07 | 83.94 | 96.07% |
| 7 | ✓ | ✓ | ✓ | 0.1~2 | 32 | 1e-4 | 1024 | Near Zero | 88.51 | 730 | 85.13 | 48.56 | 2247.40 | 82.89 | 94.38% |
| 8 | ✓ | ✓ | ✓ | 0.1~2 | 16 | 5e-5 | 1024 | Near Zero | 89.95 | 756 | 85.57 | 49.67 | 2303.66 | 83.56 | 96.13% |
| 9 | ✓ | ✓ | ✓ | 0.1~2 | 16 | 5e-5 | 1792 | Random | 89.36 | 777 | 85.42 | 48.56 | 2311.13 | 84.92 | 96.36% |
| 10 | ✓ | ✓ | ✓ | 0.1~2 | 16 | 5e-5 | 1792 | Kaiming | 89.61 | 761 | 85.52 | 49.00 | 2311.41 | 84.83 | 96.23% |
| 11(VisionSelector) | ✓ | ✓ | ✓ | 0.1~2 | 16 | 5e-5 | 1792 | Near Zero | 89.91 | 770 | 85.57 | 49.22 | 2293.54 | 84.27 | 96.33% |

## A.4 ANALYSIS OF MODEL PERFORMANCE UNDER DIFFERENT COMPRESSION BUDGETS

We analyze model performance under different compression budgets. Figure 4 plots the average performance retention rate across the nine image understanding datasets detailed in Table 1, comparing models trained and evaluated under different compression budgets. The analysis reveals three key findings: First, a model performs robustly when the testing compression budget is equal to or higher than its training budget. This shows that our method has learned the most critical tokens under different training compression budgets. Second, performance degrades significantly when a model is tested at a compression rate stricter than its training condition. This implies that a loose training compression rate weakens the model's ability to discriminate among critical visual tokens. Third, training with a moderate compression rate (e.g., 20%) enhances overall robustness. The model trained at a 20% budget maintains high performance across most tested budgets. Its consistently strong performance across all budgets makes it more versatile for practical deployments where testing constraints may vary.

Although its performance fluctuates with the training budget, our method's overall effectiveness surpasses most heuristic-based approaches, highlighting the superiority of a learnable compression strategy over rule-based ones.

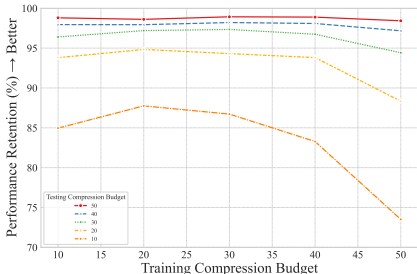

Figure 4: The impact of training and testing compression budgets on performance retention. The plot illustrates the model's performance robustness when trained under one compression budget (x-axis) and evaluated under another (each colored line). A key observation is that models generalize well to less strict compression budgets but suffer a significant performance degradation when subjected to stricter compression than they were trained on.

## A.5 CORNER CASE ANALYSIS AND VISUALIZATION

To complement the quantitative results, we conduct a qualitative analysis on the POPE dataset that focuses on two aspects: (1) robustness under extreme pruning where rare-but-critical tokens may be at risk, and (2) failure modes in visually complex scenes where the token budget is very limited.

### A.5.1 ROBUSTNESS ON RARE-BUT-CRITICAL TOKENS AND BIAS CHECK

A key concern in hard token selection is the potential removal of small yet semantically critical regions, as well as systematic bias against specific semantic categories. We visualize selection masks under pruning budgets of 10%, 5%, and 2.5%.

As shown in Figure 5, VisionSelector consistently preserves the most informative regions even under very aggressive pruning. For example, the skier in the snow scene (Row 1) and the kites in the sky (Row 2) remain selected although they occupy only a few tokens. This behavior is consistent with the end-to-end training objective. During training, discarding such critical tokens would incur a large task loss, which in turn produces strong gradients that push their importance scores up, so at inference time these regions tend to be preserved. The visualization also indicates no observable semantic bias. Foreground regions containing humans (Row 1 and Row 4), objects such as furniture and equipment (Row 2 and Row 3), and animals (Row 4) are consistently retained, while mostly redundant background areas such as snow, sky, walls, and grass are pruned.

### A.5.2 FAILURE MODES AND INFORMATION BOTTLENECK IN COMPLEX SCENES

We further analyze cases where the model fails to retain all semantic content. We compare masks obtained with pruning budgets of 30%, 20%, and 10% in visually cluttered scenes.

Figure 6 shows that with a 30% budget VisionSelector covers most salient objects even in crowded urban scenes and sports scenes. When the budget decreases to 20% and 10%, the mask becomes sparse and some objects or object parts are no longer covered. For example, in the street scene in Row 2 the 10% mask keeps the vehicle but misses parts of the surrounding people. This behavior is not consistent with a semantic bias toward or against particular categories. Instead it reflects an information bottleneck. In high-entropy scenes with many distinct regions, a very small token budget is insufficient to cover all informative areas, so the model is forced to retain only the most discriminative subset. This observation suggests that fixed global pruning ratios are suboptimal and motivates future work on adaptive budgets or near-lossless compression strategies that depend on scene complexity.

## A.6 ADDITIONAL EXPERIMENTS ON QWEN2.5-VL-3B

**Implementation Details** We utilize the Qwen2.5-VL-3B (Bai et al., 2025) model as its foundation. We perform all training and inference on a platform consisting of 8 NVIDIA A800 (80G) GPUs. The consolidated training data includes the OCRVQA, ChartQA, and a randomly sampled 10% subset

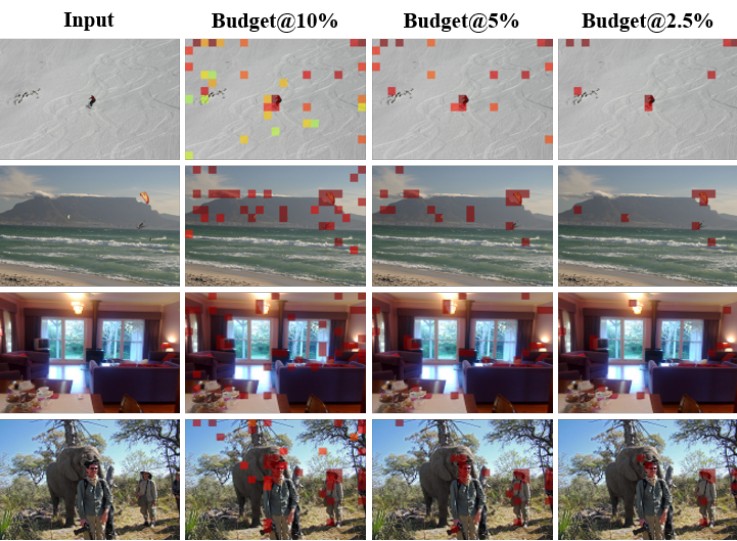

Figure 5: Visualization on POPE of token selection under extreme pruning budgets of 10%, 5%, and 2.5%. VisionSelector preserves rare-but-critical tokens such as the tiny skier in Row 1 and the distant kites in Row 2 even when 97.5% of tokens are removed. The selections do not exhibit systematic semantic bias and consistently focus on humans, objects, and animals in the foreground area while discarding redundant background.

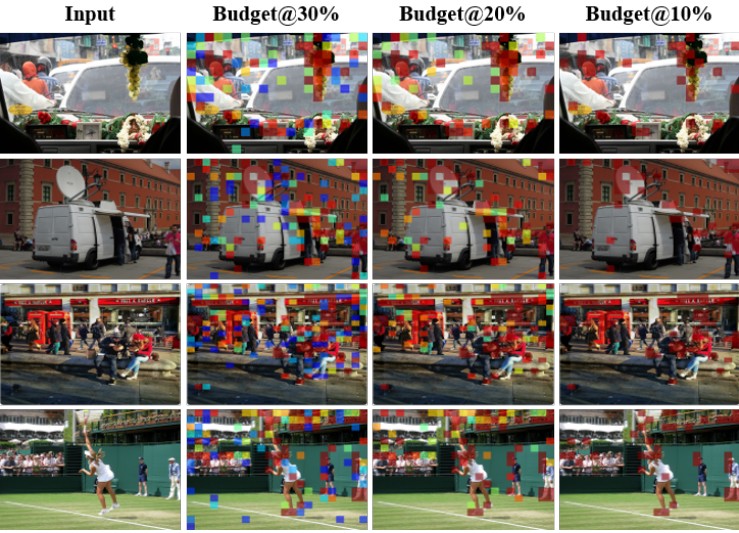

Figure 6: Failure case visualization on POPE under decreasing token budgets of 30%, 20%, and 10%. At 30% the selected tokens cover most semantic regions in complex scenes, whereas at 10% some objects and object parts are missed. This behavior reflects an information bottleneck caused by the strict budget in high-entropy images, rather than a systematic semantic bias against specific categories.

of the COCO dataset, creating a total of 144K training examples. Regarding hyperparameters, the hidden layer dimension of Qwen2.5-VL-3B is $2048$, so we define our parameter $d$ as $1024$. The learning rate is set to $5e - 5$ with a batchsize of 8. We also define $\lambda_{start} = 0.1, \lambda_{end} = 2$, and $b = 20\%$. The total number of trainable parameters in our experiment is 4.00M.

**Quantitative Results.** Table 6 presents the quantitative results of our experiments on Qwen2.5-VL-3B. Our proposed method demonstrates superior performance over state-of-the-art approaches,

including the learnable Dynamic method, across the majority of datasets. Furthermore, it achieves a competitive result on the ScienceQA dataset.

Table 6: Comparison results of our method and different baselines on image-language understanding datasets under Qwen2.5-VL-3B.

| Method | DocVQA Anls | ChartQA Relaxed | TextVQA EM | OCRBench Acc | ScienceQA EM | AI2D EM | MMMU Acc | MME Score | POPE F1 | Avg |
|---|---|---|---|---|---|---|---|---|---|---|
| *Dynamic Resolution(MinPix=256×28×28,MaxPix=2048×28×28),Upper Bound (100%)* | | | | | | | | | | |
| Avg. Visual Tokens | 1951.61 | 596.06 | 976.58 | 652.82 | 323.05 | 510.19 | 601.15 | 867.67 | 359.55 | |
| Qwen-2.5-VL-3B | 92.76 | 83.40 | 77.87 | 788 | 80.37 | 90.84 | 46.78 | 2168.46 | 86.94 | 100% |
| *Retain 30% Tokens (70% Compression Ratio)* | | | | | | | | | | |
| FastV (ECCV2024) | 81.66 88.03% | 70.04 83.98% | 74.02 95.06% | 629 79.82% | 78.63 97.96% | 86.40 95.11% | 46.00 98.33% | 2048.53 94.47% | 82.94 95.40% | 92.02% |
| PruMerge+ (ICCV2025) | 65.43 70.54% | 63.88 76.59% | 64.29 82.56% | 553 70.18% | 78.93 98.33% | 79.24 87.23% | 46.00 98.33% | 2051.01 94.58% | 84.78 97.52% | 86.21% |
| VisionZip (CVPR2025) | 81.38 87.73% | 75.40 90.41% | 67.71 86.95% | 635 80.58% | 79.47 99.00% | 84.88 93.44% | 45.44 97.14% | 2065.69 95.26% | **86.00** 98.92% | 92.16% |
| DART (EMNLP2025) | 65.96 71.11% | 64.80 77.70% | 63.64 81.73% | 541 68.65% | 79.72 99.31% | 72.28 79.57% | 45.44 97.14% | 1996.75 92.08% | 84.85 97.60% | 84.99% |
| DivPrune (CVPR2025) | 73.99 79.76% | 66.48 79.71% | 70.34 90.33% | 631 80.08% | 78.18 97.40% | 83.42 91.83% | 45.44 97.14% | 2010.72 92.73% | 86.38 99.36% | 89.81% |
| Dynamic (ICLR2025) | 85.28 91.94% | 75.4 90.41% | 75.63 97.12% | 738 93.65% | 79.67 99.13% | 87.82 96.68% | 45.22 96.67% | 2078.06 95.83% | 85.13 97.92% | 95.48% |
| **VisionSelector** | **88.87** 95.81% | **75.60** 90.65% | **76.28** 97.96% | 752 95.43% | 80.22 99.94% | **88.63** 97.57% | **46.11** 98.57% | **2095.06** 96.62% | 85.95 98.86% | **96.28%** |
| *Retain 20% Tokens (80% Compression Ratio)* | | | | | | | | | | |
| FastV (ECCV2024) | 72.90 78.59% | 62.00 74.34% | 71.41 91.70% | 560 71.07% | 78.88 98.27% | 83.29 91.69% | 45.11 96.43% | 1992.58 91.89% | 80.01 92.03% | 87.33% |
| PruMerge+ (ICCV2025) | 52.58 56.68% | 54.76 65.66% | 58.44 75.05% | 484 61.42% | 79.97 99.63% | 74.06 81.53% | 44.89 95.96% | 1977.90 91.21% | 83.23 95.73% | 80.32% |
| VisionZip (CVPR2025) | 69.62 75.05% | 65.76 78.85% | 59.97 77.01% | 506 64.21% | 79.97 99.63% | 78.79 86.73% | 45.67 97.63% | 1892.24 87.26% | 84.17 96.81% | 84.80% |
| DART (EMNLP2025) | 51.01 54.99% | 54.24 65.04% | 56.44 72.48% | 537 68.15% | **80.96** 100.86% | 69.07 76.03% | 45.33 96.90% | 1925.04 88.77% | 81.93 94.24% | 79.72% |
| DivPrune (CVPR2025) | 63.82 68.80% | 57.64 69.11% | 66.76 85.73% | 535 67.89% | 80.46 100.24% | 77.59 85.41% | 45.56 97.39% | 1955.53 90.18% | **85.47** 98.31% | 84.79% |
| Dynamic (ICLR2025) | 76.47 82.44% | 70.2 84.17% | 74.42 95.57% | 672 85.28% | 78.83 98.08% | 85.49 94.11% | 44.33 94.76% | 2053.26 94.69% | 83.34 95.86% | 91.66% |
| **VisionSelector** | **84.32** 90.90% | **73.48** 88.11% | **74.97** 96.28% | 705 89.47% | 79.82 99.44% | 86.59 95.32% | **46.56** 99.53% | **2055.00** 94.77% | 84.87 97.62% | **94.60%** |
| *Retain 10% Tokens (90% Compression Ratio)* | | | | | | | | | | |
| FastV (ECCV2024) | 54.62 58.88% | 46.96 56.31% | 65.58 84.22% | 411 52.16% | 79.33 98.83% | 77.07 84.84% | **45.11** 96.43% | 1815.70 83.73% | 70.28 80.84% | 77.36% |
| PruMerge+ (ICCV2025) | 37.51 40.44% | 46.68 55.97% | 46.43 59.63% | 350 44.42% | 78.93 98.33% | 68.43 75.33% | 44.56 95.25% | 1777.60 81.98% | 77.55 89.20% | 71.17% |
| VisionZip (CVPR2025) | 41.67 44.92% | 46.84 56.16% | 43.83 56.29% | 321 40.74% | 79.47 99.00% | 69.43 76.43% | 45.22 96.67% | 1725.49 79.57% | 78.60 90.41% | 71.13% |
| DART (EMNLP2025) | 32.32 34.84% | 39.44 47.29% | 44.39 57.01% | 315 39.97% | 79.72 99.31% | 65.54 72.15% | 44.33 94.76% | 1733.77 79.95% | 75.41 86.74% | 68.00% |
| DivPrune (CVPR2025) | 47.13 50.81% | 44.36 53.19% | 57.80 74.23% | 394 50.00% | 78.19 97.41% | 70.14 77.21% | 44.67 95.49% | 1782.737 82.21% | 80.50 92.59% | 74.79% |
| Dynamic (ICLR2025) | 58.28 62.83% | 61.56 73.81% | 70.14 90.07% | 556 70.56% | 79.08 98.39% | 78.72 86.66% | 44.33 94.76% | 1887.79 87.06% | 75.01 86.28% | 83.38% |
| **VisionSelector** | **68.36** 73.70% | **65.04** 77.99% | **69.92** 89.79% | 587 74.49% | **80.12** 99.81% | 81.09 89.27% | 44.67 95.49% | **1928.75** 88.95% | 81.72 94.00% | **87.05%** |

## A.7 ADDITIONAL EXPERIMENTS ON QWEN2.5-VL-32B

To verify the effectiveness of our method on a larger scale, we evaluate VisionSelector on Qwen2.5-VL-32B (Bai et al., 2025).

**Implementation Details** All training and inference tasks are conducted on a platform equipped with 8 NVIDIA A800 (80G) GPUs. The consolidated training dataset consists of 144K examples derived from OCRVQA, ChartQA, and a 10% random subset of COCO. Given the hidden dimension of 5120 in the 32B model, we set the parameter $d$ to 2560. The training configuration employs a learning rate of $5e-5$ and a batch size of 16. We further initialize $\lambda_{start}$ at 0.1 and $\lambda_{end}$ at 2, with the budget $b$ fixed at 20%.

**Quantitative Results** Table 7 presents the quantitative results of our experiments on Qwen2.5-VL-32B. Our proposed method demonstrates superior performance over state-of-the-art approaches across the majority of datasets. Furthermore, it achieves a competitive result on the MME dataset.

Table 7: Comparison results of our method and different baselines on image-language understanding datasets under Qwen2.5-VL-32B.

| Method | DocVQA Anls | ChartQA Relaxed | TextVQA EM | OCRBench Acc | ScienceQA EM | AI2D EM | MMMU Acc | MME Score | Avg |
|---|---|---|---|---|---|---|---|---|---|
| *Dynamic Resolution(MinPix=256×28×28,MaxPix=2048×28×28),Upper Bound (100%)* | | | | | | | | | |
| Avg. Visual Tokens | 1951.61 | 596.06 | 976.58 | 652.82 | 323.05 | 510.19 | 601.15 | 867.67 | |
| Qwen-2.5-VL-32B | 91.10 | 64.20 | 66.76 | 649 | 91.62 | 94.20 | 60.11 | 2450.12 | 100% |
| *Retain 20% Tokens (80% Compression Ratio)* | | | | | | | | | |
| FastV (ECCV2024) | 54.89 / 60.25% | 36.24 / 56.45% | 53.18 / 79.66% | 371 / 57.16% | 81.80 / 89.28% | 82.41 / 87.48% | 55.78 / 92.80% | 1990.81 / 81.25% | 75.54% |
| PruMerge+ (ICCV2025) | *oom* / / | 33.00 / 51.40% | 48.96 / 73.34% | 357 / 55.01% | 87.11 / 95.08% | 76.61 / 81.33% | *oom* / / | 2086.12 / 85.14% | / |
| VisionZip (CVPR2025) | 60.76 / 66.70% | 46.32 / 72.15% | 50.10 / 75.04% | 341 / 52.54% | 85.62 / 93.45% | 80.21 / 85.15% | *oom* / / | 2094.90 / 85.50% | / |
| DART (EMNLP2025) | 51.22 / 56.22% | 35.24 / 54.89% | 57.91 / 86.74% | 443 / 68.26% | 85.37 / 93.18% | 74.29 / 78.86% | 56.22 / 93.53% | 2178.25 / 88.90% | 77.57% |
| DivPrune (CVPR2025) | 65.27 / 71.65% | 36.92 / 57.51% | 58.35 / 87.40% | 432 / 66.56% | 85.03 / 92.81% | 81.70 / 86.73% | 57.67 / 95.94% | **2208.52** / 90.14% | 81.09% |
| **VisionSelector** | **74.46** / 81.73% | **48.16** / 75.02% | **64.17** / 96.12% | **543** / 83.67% | **87.90** / 95.94% | **89.09** / 94.58% | **59.00** / 98.15% | 2197.50 / 89.69% | **89.36%** |
| *Retain 10% Tokens (90% Compression Ratio)* | | | | | | | | | |
| FastV (ECCV2024) | 38.22 / 41.95% | 21.96 / 34.21% | 41.49 / 62.15% | 235 / 36.21% | 81.21 / 88.64% | 75.58 / 80.23% | 54.00 / 89.84% | 1752.91 / 71.54% | 63.10% |
| PruMerge+ (ICCV2025) | *oom* / / | 27.32 / 42.55% | 37.45 / 56.10% | 224 / 34.51% | 82.94 / 90.53% | 69.69 / 73.98% | *oom* / / | 1832.58 / 74.80% | / |
| VisionZip (CVPR2025) | 35.76 / 39.25% | 28.80 / 44.86% | 36.38 / 54.49% | 242 / 37.29% | 82.80 / 90.37% | 71.5 / 75.90% | *oom* / / | 1929.95 / 78.77% | / |
| DART (EMNLP2025) | 33.40 / 36.66% | 24.88 / 38.75% | 47.83 / 71.64% | 342 / 52.70% | 81.06 / 88.47% | 68.52 / 72.74% | 54.67 / 90.95% | 2029.26 / 82.82% | 66.84% |
| DivPrune (CVPR2025) | 48.01 / 52.70% | 25.40 / 39.56% | 51.00 / 76.39% | 310 / 47.77% | 81.31 / 88.75% | 71.92 / 76.35% | 54.78 / 91.13% | **2072.38** / 84.58% | 69.65% |
| **VisionSelector** | **53.1** / 58.29% | **41.96** / 65.36% | **59.66** / 89.36% | **413** / 63.64% | **83.69** / 91.34% | **81.02** / 86.01% | **55.33** / 92.05% | 2007.73 / 81.94% | **78.50%** |

## A.8 ADDITIONAL EXPERIMENTS ON LLAVA-ONEVISION-1.5-8B

To evaluate the effectiveness of our method across different model architectures, we conduct additional experiments on LLaVA-OneVision-1.5-8B (An et al., 2025). We use a batch size of 16, a learning rate of $5e - 5$, and set the hyperparameters to $\lambda_{start} = 0.1$ and $\lambda_{end} = 3$. The projection dimension $d$ for $W_q$ and $W_k$ is set to 2048. The retention budget for visual tokens is set to 20%. For the constraint loss weight, $\lambda_t$, we adopt a Curriculum Annealing Strategy, linearly increasing it from an initial value of 0.1 to a final value of 3.0 over the course of training. The model trains for 1 epoch on 8 NVIDIA A800 (80GB) GPUs, which takes about 2 hours. The number of trainable parameters is 16.87M, accounting for 0.20% of the total parameters. We evaluate our method on 9 image-language understanding benchmarks, as summarized in Table 8.

The results show that the performance of FastV on LLaVA-OneVision-1.5-8B is relatively lower compared with its results on Qwen2.5-VL, suggesting that training-free methods are influenced by the internal feature distribution of the underlying model and may exhibit varying behaviors across different architectures. VisionZip encounters CUDA out-of-memory (OOM) issues on high-resolution datasets such as DocVQA and multi-image datasets such as MMMU, which is likely related to the specific layers where its attention operations are applied. This observation indicates that further optimization may be needed to enhance its scalability in real-world deployment. DivPrune achieves competitive results in general scenarios, while its performance on fine-grained semantic understanding tasks (e.g., DocVQA, ChartQA, and OCRBench) is relatively reduced, likely because the diversity preservation process filters out some fine-grained semantic tokens.

Table 8: Comparison results of our method and different baselines on image-language understanding datasets under LLaVA-OneVision-1.5-8B.

| Method | DocVQA Anls | ChartQA Relaxed | TextVQA EM | OCRBench Acc | ScienceQA EM | AI2D EM | MMMU Acc | MME Score | Avg |
|---|---|---|---|---|---|---|---|---|---|
| *Dynamic Resolution(MinPix=256×28×28,MaxPix=2048×28×28),Upper Bound (100%)* | | | | | | | | | |
| Avg. Visual Tokens | 3795.45 | 595.92 | 978.56 | 830.65 | 320.4 | 523.27 | 670.11 | 1154.48 | |
| LLaVA-OneVision-1.5-8B | 97.87 | 86.72 | 79.51 | 830 | 98.56 | 94.01 | 56.44 | 2271.32 | 100% |
| *Retain 30% Tokens (70% Compression Ratio)* | | | | | | | | | |
| FastV (ECCV2024) | 86.51 88.39% | 68.64 79.15% | 72.27 90.89% | 596 71.81% | 91.22 92.55% | 83.52 88.84% | 53.67 95.09% | 2019.84 88.93% | 86.96% |
| VisionZip (CVPR2025) | *oom* / | 78.04 89.99% | 73.25 92.13% | *oom* / | 96.23 97.64% | 85.33 90.77% | *oom* / | *oom* / | / |
| DivPrune (CVPR2025) | 85.23 87.08% | 70.64 81.46% | 76.03 95.62% | 645 77.71% | 94.65 96.03% | 89.35 95.04% | 54.11 95.87% | 2104.77 92.67% | 90.18% |
| **VisionSelector** | **95.46** 97.54% | **79.68** 91.88% | **78.12** 98.25% | **755** 90.96% | **97.27** 98.69% | **91.77** 97.62% | **55.00** 97.45% | **2241.82** 98.70% | **96.39%** |
| *Retain 20% Tokens (80% Compression Ratio)* | | | | | | | | | |
| FastV (ECCV2024) | 76.64 78.31% | 59.08 68.13% | 67.33 84.68% | 489 58.92% | 87.95 89.23% | 78.89 83.92% | 51.33 90.95% | 1918.66 84.47% | 79.83% |
| VisionZip (CVPR2025) | oom / | 67.56 77.91% | 66.28 83.36% | oom / | 91.87 93.21% | 79.18 84.23% | oom / | oom / | / |
| DivPrune (CVPR2025) | 75.90 77.55% | 58.64 67.62% | 72.73 91.47% | 556 66.99% | 92.46 93.81% | 84.26 89.63% | 52.56 93.13% | 2057.83 90.60% | 83.85% |
| **VisionSelector** | **90.51** 92.48% | **74.72** 86.16% | **76.02** 95.61% | **677** 81.57% | **95.79** 97.19% | **90.45** 96.21% | **55.11** 97.64% | **2182.84** 96.10% | **92.86%** |
| *Retain 10% Tokens (90% Compression Ratio)* | | | | | | | | | |
| FastV (ECCV2024) | 54.43 55.61% | 38.08 43.91% | 56.47 71.02% | 374 45.06% | 81.51 82.70% | 73.15 77.81% | 49.56 87.81% | 1800.00 79.25% | 67.89% |
| VisionZip (CVPR2025) | *oom* / | 43.12 49.72% | 47.17 59.33% | *oom* / | 88.70 90.00% | 71.83 76.41% | *oom* / | *oom* / | / |
| DivPrune (CVPR2025) | 56.48 57.71% | 38.24 44.10% | 65.48 82.35% | 419 50.48% | 88.20 89.49% | 76.85 81.75% | **51.33** 90.95% | 1897.29 83.53% | 72.54% |
| **VisionSelector** | **71.92** 73.49% | **62.12** 71.63% | **67.40** 84.77% | **456** 54.94% | **89.89** 91.20% | **84.29** 89.66% | **51.33** 90.95% | **2002.44** 88.16% | **80.60%** |

## A.9 THEORETICAL ANALYSIS OF THE DIFFERENTIABLE TOP-K OPERATOR

We provide theoretical insights into the differentiable Top-K operator in VisionSelector, highlighting its key properties: (i) monotonicity, (ii) shift invariance, (iii) gradient stability, (iv) proof of gradient maximization at the decision boundary, (v) backpropagation and constraint satisfaction, and (vi) optimization dynamics for token retention.

Specifically, the differentiable Top-K operator is defined as:

$$\mathbf{M} = \text{DiffTopK}(\mathbf{x}) = \sigma(\mathbf{x} + t), \tag{16}$$

where $\sigma(z) = (1 + e^{-z})^{-1}$ denotes the standard Sigmoid function. The threshold parameter $t$ is dynamically adjusted to satisfy the **cardinality constraint**:

$$\sum_{i=1}^{N} \sigma(x_i + t) = k, \tag{17}$$

ensuring that the expected number of selected tokens equals the target number $k$.

### A.9.1 MONOTONICITY

The differentiable Top-K operator preserves the order of token importance scores, i.e., it is *monotonic* with respect to the input $\mathbf{x}$. For any two tokens $i$ and $j$, if $x_i > x_j$, then their corresponding selection probabilities satisfy

$$M_i > M_j. \tag{18}$$

To see this, note that $M_i = \sigma(x_i + t)$, where $\sigma(x) = (1 + e^{-x})^{-1}$. Hence, we analyze its derivative:

$$\frac{d}{dx}\sigma(x) = \frac{d}{dx}(1 + e^{-x})^{-1} = \frac{e^{-x}}{(1 + e^{-x})^2}. \tag{19}$$

This can be rewritten as $\sigma'(x) = \sigma(x)(1 - \sigma(x))$. Since the range of the Sigmoid function is $(0, 1)$, both $\sigma(x)$ and $(1 - \sigma(x))$ are strictly positive. Therefore, $\sigma'(x) > 0$ for all $x \in \mathbb{R}$, proving that the Sigmoid function is strictly monotonically increasing.

Given $x_i > x_j$, it follows that $x_i + t > x_j + t$. Applying the strictly increasing Sigmoid function, we get:

$$\sigma(x_i + t) > \sigma(x_j + t), \tag{20}$$

which implies $M_i > M_j$. Therefore, the ranking structure of $\mathbf{x}$ is preserved under the continuous relaxation $M_i$, and the soft operator provides an order-consistent approximation of discrete Top-K, an essential property for differentiable ranking consistency.

### A.9.2 SHIFT INVARIANCE

The differentiable Top-K operator is invariant under additive shifts to the input scores, provided the threshold $t$ adapts to satisfy the cardinality constraint. Formally, for any constant shift $c \in \mathbb{R}$, let the shifted input be $\tilde{\mathbf{x}} = \mathbf{x} + c$. We show that the resulting selection probabilities remain unchanged:

$$M_i(\mathbf{x} + c) = M_i(\mathbf{x}). \tag{21}$$

Recall that the selection probability is defined as $M_i = \sigma(x_i + t)$, subject to the constraint $\sum_{i=1}^{N} \sigma(x_i + t) \approx k$. This implies that $t$ is implicitly determined by the input $\mathbf{x}$ to satisfy this sum. Let $t^*$ be the threshold that satisfies the constraint for the original input $\mathbf{x}$:

$$\sum_{i=1}^{N} \sigma(x_i + t^*) = k. \tag{22}$$

Now, consider the shifted input $\tilde{x}_i = x_i + c$. We seek a new threshold $\tilde{t}$ such that the constraint holds:

$$\sum_{i=1}^{N} \sigma(\tilde{x}_i + \tilde{t}) = \sum_{i=1}^{N} \sigma(x_i + c + \tilde{t}) = k. \tag{23}$$

Comparing the two equations, since the function $f(t) = \sum \sigma(x_i + t)$ is strictly monotonic, the argument inside the Sigmoid must be effectively identical to satisfy the same sum $k$. Thus, we must have:

$$c + \tilde{t} = t^* \implies \tilde{t} = t^* - c. \tag{24}$$

Substituting $\tilde{t}$ back into the probability formula for the shifted input:

$$\tilde{M}_i = \sigma(\tilde{x}_i + \tilde{t}) = \sigma((x_i + c) + (t^* - c)) = \sigma(x_i + t^*) = M_i. \tag{25}$$

This demonstrates that the adaptive threshold $\tilde{t}$ automatically compensates for the input shift $c$, rendering the final selection probabilities $M_i$ invariant to translation.

### A.9.3 GRADIENT STABILITY

The differentiable operator provides stable gradients for end-to-end training. To analyze this, we examine the Jacobian of the selection vector $\mathbf{M}$ with respect to the input scores $\mathbf{x}$. Since the threshold $t$ is determined by the cardinality constraint $\sum_{i=1}^{N} \sigma(x_i + t) = k$, $t$ is an implicit function of $\mathbf{x}$, which we denote as $t(\mathbf{x})$.

The derivative of the selection probability $M_i = \sigma(x_i + t(\mathbf{x}))$ with respect to an input score $x_j$ is given by the chain rule:

$$\frac{\partial M_i}{\partial x_j} = \frac{\partial \sigma(x_i + t)}{\partial x_j} = \sigma'(x_i + t) \cdot \left( \frac{\partial x_i}{\partial x_j} + \frac{\partial t}{\partial x_j} \right). \tag{26}$$

Here, $\frac{\partial x_i}{\partial x_j}$ is the Kronecker delta, $\delta_{ij}$ (which is 1 if $i = j$ and 0 otherwise). Let's denote $\sigma'_i = \sigma'(x_i + t)$. The equation becomes:

$$\frac{\partial M_i}{\partial x_j} = \sigma'_i \left( \delta_{ij} + \frac{\partial t}{\partial x_j} \right). \tag{27}$$

To find the term $\frac{\partial t}{\partial x_j}$, we differentiate the constraint equation $\sum_{l=1}^{N} \sigma(x_l + t) = k$ with respect to $x_j$:

$$\frac{\partial}{\partial x_j} \sum_{l=1}^{N} \sigma(x_l + t) = \sum_{l=1}^{N} \left[ \sigma'_l \cdot \left( \delta_{lj} + \frac{\partial t}{\partial x_j} \right) \right] = 0. \tag{28}$$

Solving for $\frac{\partial t}{\partial x_j}$:

$$\sigma'_j + \frac{\partial t}{\partial x_j} \sum_{l=1}^{N} \sigma'_l = 0 \implies \frac{\partial t}{\partial x_j} = -\frac{\sigma'_j}{\sum_{l=1}^{N} \sigma'_l}. \tag{29}$$

Substituting this back into Eq. equation 27, we get the full Jacobian entry:

$$\frac{\partial M_i}{\partial x_j} = \sigma'_i \left( \delta_{ij} - \frac{\sigma'_j}{\sum_{l=1}^{N} \sigma'_l} \right). \tag{30}$$

The derivative of the Sigmoid function, $\sigma'(z) = \sigma(z)(1 - \sigma(z))$, has a maximum value of $1/4$ (at $z = 0$). This means both $\sigma'_i$ and $\sigma'_j$ are bounded by $1/4$. The term $\sum_{l=1}^{N} \sigma'_l$ is a sum of $N$ non-negative, bounded values. As long as at least one score $x_l$ is near the threshold $-t$, this sum is non-zero, preventing the gradient from becoming undefined.

Crucially, the gradient magnitude does not depend on the scale of the input scores $\mathbf{x}$, but rather on how close the scores are to the decision boundary $-t$. This structure ensures that gradients are well-behaved and do not explode, which is a key advantage over sorting-based relaxation methods that can suffer from vanishing or exploding gradients(Zhu et al., 2025). This bounded-gradient property allows for a stable soft-to-hard transition during training, enabling robust optimization.

### A.9.4 PROOF OF GRADIENT MAXIMIZATION AT THE DECISION BOUNDARY

We analyze the magnitude of the Jacobian entries and prove that they are maximized when the selection probabilities are closest to the decision boundary (i.e., $M_i \approx 0.5$).

Recall the Jacobian derived in Eq. equation 30:

$$J_{ij} = \frac{\partial M_i}{\partial x_j} = \sigma'_i \left( \delta_{ij} - \frac{\sigma'_j}{\sum_{l=1}^{N} \sigma'_l} \right), \tag{31}$$

where $\sigma'_k = \sigma(x_k + t)(1 - \sigma(x_k + t))$. The function $h(z) = \sigma(z)(1 - \sigma(z))$ achieves its global maximum of $0.25$ when $\sigma(z) = 0.5$ (i.e., $z = 0$). Conversely, $h(z) \to 0$ as $\sigma(z) \to 0$ or $1$. Thus, the term $\sigma'_k$ can be viewed as an "uncertainty indicator," peaking when the token is at the decision boundary.

We analyze the gradient magnitude in two cases:

**Case 1: Self-Influence (Diagonal terms, $i = j$).** This term represents how the score of token $i$ affects its own selection probability.

$$J_{ii} = \sigma'_i \left( 1 - \frac{\sigma'_i}{\sum_{l=1}^{N} \sigma'_l} \right). \tag{32}$$

Let $R_i = \sum_{l \neq i} \sigma'_l$ be the sum of the derivative terms of all other tokens. Since $\sigma'_l \geq 0$, we have $R_i \geq 0$. The expression becomes:

$$J_{ii} = \sigma'_i \left( 1 - \frac{\sigma'_i}{\sigma'_i + R_i} \right) = \sigma'_i \left( \frac{R_i}{\sigma'_i + R_i} \right) = \frac{R_i \cdot \sigma'_i}{R_i + \sigma'_i}. \tag{33}$$

Consider $J_{ii}$ as a function of the variable $u = \sigma'_i$, denoted as $g(u) = \frac{R_i u}{R_i + u}$, where $u \in (0, 0.25]$. Taking the derivative with respect to $u$:

$$g'(u) = \frac{R_i(R_i + u) - R_i u}{(R_i + u)^2} = \frac{R_i^2}{(R_i + u)^2}. \tag{34}$$

Assuming $R_i > 0$ (which holds if there is at least one other uncertain token), then $g'(u) > 0$. This implies that $J_{ii}$ is a strictly monotonically increasing function of $\sigma_i'$. Since $\sigma_i'$ is maximized when $\sigma(x_i + t) = 0.5$, the gradient $J_{ii}$ is also maximized at this point.

**Case 2: Cross-Influence (Off-diagonal terms, $i \neq j$).** This term represents how the score of token $j$ affects the probability of token $i$.

$$|J_{ij}| = \left| -\frac{\sigma_i' \sigma_j'}{\sum_{l=1}^{N} \sigma_l'} \right| = \frac{\sigma_i' \sigma_j'}{\sigma_i' + \sigma_j' + R_{ij}}, \tag{35}$$

where $R_{ij} = \sum_{l \neq i,j} \sigma_l' \geq 0$. Let $g(u, v) = \frac{uv}{u+v+R_{ij}}$ where $u = \sigma_i', v = \sigma_j'$. Taking partial derivatives:

$$\frac{\partial g}{\partial u} = \frac{v(u + v + R_{ij}) - uv}{(u + v + R_{ij})^2} = \frac{v^2 + vR_{ij}}{(u + v + R_{ij})^2} > 0. \tag{36}$$

By symmetry, $\frac{\partial g}{\partial v} > 0$. Thus, the magnitude of the cross-gradient is maximized when both $\sigma_i'$ and $\sigma_j'$ are maximized, i.e., when both tokens are near the decision boundary (0.5).

The gradients are theoretically proven to be strongest for tokens with probabilities near 0.5. This confirms that the differentiable operator naturally focuses the optimization effort on refining the "hard-to-classify" tokens at the decision boundary, rather than wasting updates on already saturated (confident) tokens.

### A.9.5 BACKPROPAGATION AND CONSTRAINT SATISFACTION

To understand how the operator updates the input scores $\mathbf{x}$ during training, we analyze the relationship between the upstream gradient $\frac{\partial L}{\partial \mathbf{M}}$ (where $L$ is the loss function) and the computed gradient $\frac{\partial L}{\partial \mathbf{x}}$. By the chain rule, the gradient with respect to the $j$-th input score $x_j$ is computed as the vector-Jacobian product:

$$\frac{\partial L}{\partial x_j} = \sum_{i=1}^{N} \frac{\partial L}{\partial M_i} \frac{\partial M_i}{\partial x_j}. \tag{37}$$

Substituting the Jacobian form derived in Eq. equation 30, we have:

$$\frac{\partial L}{\partial x_j} = \sum_{i=1}^{N} \frac{\partial L}{\partial M_i} \left[ \sigma_i' \left( \delta_{ij} - \frac{\sigma_j'}{\sum_{l=1}^{N} \sigma_l'} \right) \right]. \tag{38}$$

We can separate this summation into two distinct terms:

$$\frac{\partial L}{\partial x_j} = \underbrace{\frac{\partial L}{\partial M_j} \sigma_j'}_{\text{Local Gradient}} - \underbrace{\left( \sum_{i=1}^{N} \frac{\partial L}{\partial M_i} \sigma_i' \right) \frac{\sigma_j'}{\sum_{l=1}^{N} \sigma_l'}}_{\text{Global Correction}}. \tag{39}$$

This formulation reveals that the gradient update consists of two mechanisms:

- **Local Gating:** The first term, $\frac{\partial L}{\partial M_j} \sigma_j'$, indicates that the upstream gradient is gated by the derivative $\sigma_j'$. As shown in the previous section, this term is maximized when $x_j$ is near the threshold (where $\sigma_j'$ peaks), effectively focusing updates on uncertain tokens.

- **Global Constraint Correction:** The second term subtracts a weighted average of the gradients from all tokens. We can rewrite the final gradient as:

$$\frac{\partial L}{\partial x_j} = \sigma_j' \left( \frac{\partial L}{\partial M_j} - \frac{\sum_{i=1}^{N} \frac{\partial L}{\partial M_i} \sigma_i'}{\sum_{l=1}^{N} \sigma_l'} \right). \tag{40}$$

The term inside the parentheses represents a "centered" gradient. By subtracting the weighted mean of the upstream gradients, the operator ensures that the total change in the output probabilities respects the cardinality constraint. This mechanism introduces a competitive dynamic: for a token to increase its selection probability ($x_j \uparrow$), it must overcome not just its own threshold, but the collective tendency of all other tokens, ensuring the fixed number $k$ is maintained.

### A.9.6 Optimization Dynamics for Token Retention

We further analyze the gradient dynamics to understand how the operator promotes the retention of task-relevant tokens. Our derivation indicates that the competitive nature of the gradient update creates a corrective pressure that favors tokens with significant contributions to loss reduction.

**Gradient Competition Analysis.** Based on the Jacobian derivation in Eq. equation 30, the gradient with respect to the input score $x_k$ incorporates a centering term:

$$\frac{\partial L}{\partial x_k} = \sigma'_k \left( \frac{\partial L}{\partial M_k} - \bar{g}_{\text{avg}} \right), \quad \text{where } \bar{g}_{\text{avg}} = \frac{\sum_l \frac{\partial L}{\partial M_l} \sigma'_l}{\sum_l \sigma'_l}. \tag{41}$$

Here, $\bar{g}_{\text{avg}}$ represents the weighted average gradient of the current token set. This formulation implies that the update direction for a specific token $k$ depends on the relative magnitude of its individual gradient $\frac{\partial L}{\partial M_k}$ compared to the group average $\bar{g}_{\text{avg}}$.

**Corrective Update Trajectory.** We consider a scenario where the model excludes a potentially informative token $i$ while retaining a less relevant token $j$. An informative token typically exhibits a large negative gradient component $\frac{\partial L}{\partial M_i}$ (indicating that its inclusion effectively reduces the loss), whereas a less relevant token $j$ has a gradient $\frac{\partial L}{\partial M_j}$ close to zero. Assuming the signal from the informative token is sufficiently strong such that $|\frac{\partial L}{\partial M_i}| > |\bar{g}_{\text{avg}}|$, we observe the following dynamics:

1. **Upward Pressure on Informative Token** $i$**:** Since $\frac{\partial L}{\partial M_i}$ is negative and dominates the average term, the difference $\left( \frac{\partial L}{\partial M_i} - \bar{g}_{\text{avg}} \right)$ remains negative. In gradient descent optimization $(x \leftarrow x - \eta \nabla x)$, this negative gradient results in a positive update to the score $x_i$. Consequently, the optimization applies upward pressure on the informative token, promoting its entry into the selected set.

2. **Downward Pressure on Less Relevant Token** $j$**:** For the less relevant token where $\frac{\partial L}{\partial M_j} \approx 0$, the gradient is dominated by the term $-\bar{g}_{\text{avg}}$. If the collective gradient average $\bar{g}_{\text{avg}}$ is negative (driven by other informative tokens in the batch), the resulting term $-\bar{g}_{\text{avg}}$ becomes positive. This positive gradient leads to a negative update for score $x_j$, thereby suppressing the score of the less relevant token.

This analysis demonstrates that the differentiable operator enforces a competitive update process. Rather than relying on heuristic selection, the operator utilizes the relative gradient magnitude to iteratively refine the subset, naturally aligning the selection with the objective of minimizing the task loss.

### A.9.7 Summary

Collectively, these theoretical analyses establish a rigorous mathematical foundation for the VisionSelector framework. The properties of **monotonicity** and **shift invariance** ensure that the soft operator preserves the ranking order of input scores and remains robust to constant additive shifts. The investigation into **gradient stability** and **maximization at the decision boundary** confirms that the operator produces bounded gradients that focus optimization efforts on refining uncertain tokens (where selection probability is near 0.5), rather than saturated ones. Furthermore, the decomposition of **backpropagation** reveals a mechanism of global constraint correction, which enforces the cardinality constraint by subtracting a weighted mean from the gradients. Finally, the analysis of **token retention** illustrates that this competitive gradient dynamic applies upward pressure to informative tokens and downward pressure to less relevant ones, thereby promoting the selection of task-essential features based on their contribution to loss reduction.

### A.10 Algorithmic Procedure of DiffTopK

Following Equation 3, Differentiable Top-K solves the sigmoid threshold $t$ during the forward pass via the method of undetermined coefficients and obtains $t$ by binary search, yielding a smooth approximation to Top-K so that each probability $M_i$ lies in $[0, 1]$. Owing to the strict monotonicity of

$\sigma(\cdot)$, its compensability under global shifts (via adjusting $t$), and its saturation behavior in the low-temperature or extreme-threshold limit, the mapping $s \to M_{soft} = \sigma(s + t)$ satisfies monotonicity, invariance to global shifts, and convergence toward standard Top-K.

During backpropagation, the threshold $t$ varies with $s$ but they satisfy an implicit constraint. Differentiating this implicit equation yields a closed-form gradient, and the resulting backward computation appears in Algorithm 1.

---

**Algorithm 1:** Differentiable Top-K

---

**Input:** $s \in \mathbb{R}^{B \times N}$ (scores), $k \in \mathbb{Z}^+$ (select count)
**Output:** $M \in [0, 1]^{B \times N}$ (soft mask)
**Function** $\texttt{FindThresh}(s, k)$
    $lower \leftarrow -\max(s, \text{ dim} = 1) - 10$
    $upper \leftarrow -\min(s, \text{ dim} = 1) + 10$
    **for** $i \leftarrow 1$ **to** $64$ **do**
        $mid \leftarrow (lower + upper)/2$
        $m\_sum \leftarrow \sum \sigma(s + mid)$
        $mask \leftarrow m\_sum < k$
        $lower[mask] \leftarrow mid[mask]$
        $upper[\neg mask] \leftarrow mid[\neg mask]$
    **return** $(lower + upper)/2$
**Procedure** *Forward* $(s, k)$
    $t \leftarrow \texttt{FindThresh}(s, k)$
    **return** $\sigma(s + t)$
**Procedure** *Backward* $(grad)$
    $v \leftarrow \sigma'(s + t)$
    $s_{sum} \leftarrow \sum v$
    $uv \leftarrow grad \odot v$
    $uv\_sum \leftarrow \sum uv$
    $grad\_s \leftarrow uv - (uv\_sum/s_{sum}) \odot v$
    **return** $grad\_s$

---

### A.11 MORE VISULIZATIONS

In Figure 7, we present examples on the MME dataset under a $30\%$ retention budget where our method outperforms Qwen2.5-VL-7B. This effect likely arises because our method guides attention: it does not discard information at random but precisely removes "noise" tokens that interfere with the model's response. This observation suggests that in MLLMs, the quality of visual tokens is more important than their quantity.

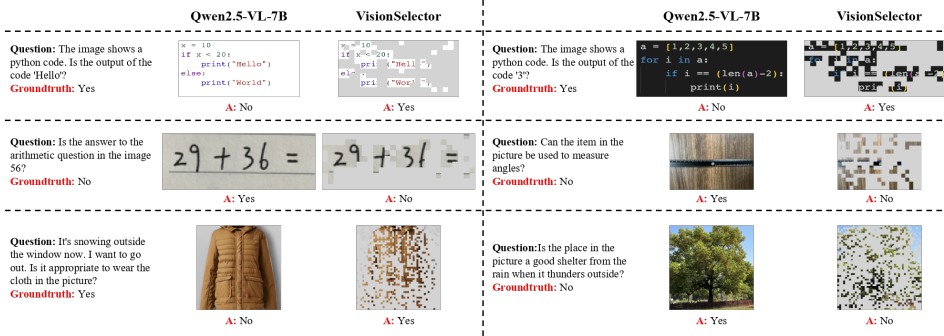

Figure 7: Qualitative examples on the MME dataset under a $30\%$ retention budget: our method preserves informative regions and removes distracting "noise" tokens, yielding clearer attention and better answers than Qwen2.5-VL-7B across code reasoning, numerical calculation, and common-sense reasoning.

And We provide additional visualizations. In Figure 8 and Figure 9, we present qualitative comparisons on the TextVQA dataset under a 20% budget. In Figure 10, we present qualitative comparisons on the OCRBench dataset under a 20% budget. The results show that our method identifies the most critical visual tokens and produces accurate answers for both natural images and charts.

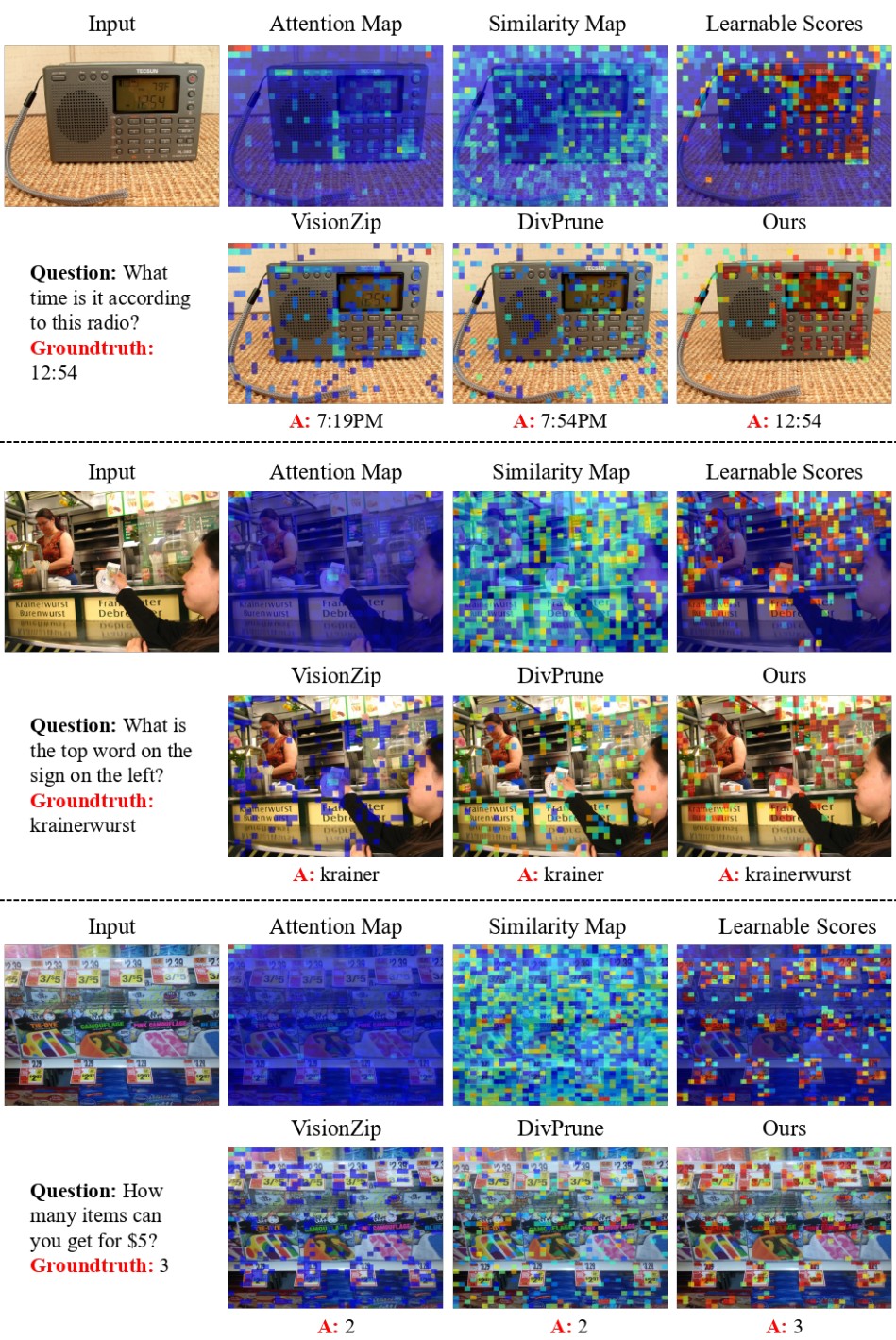

Figure 8: More qualitative comparison of VisionSelector on TextVQA. The top row visualizes different token scoring criteria: standard attention, a similarity map, and our learned importance scores. The bottom row shows the resulting token selections and model predictions at a 20% budget.

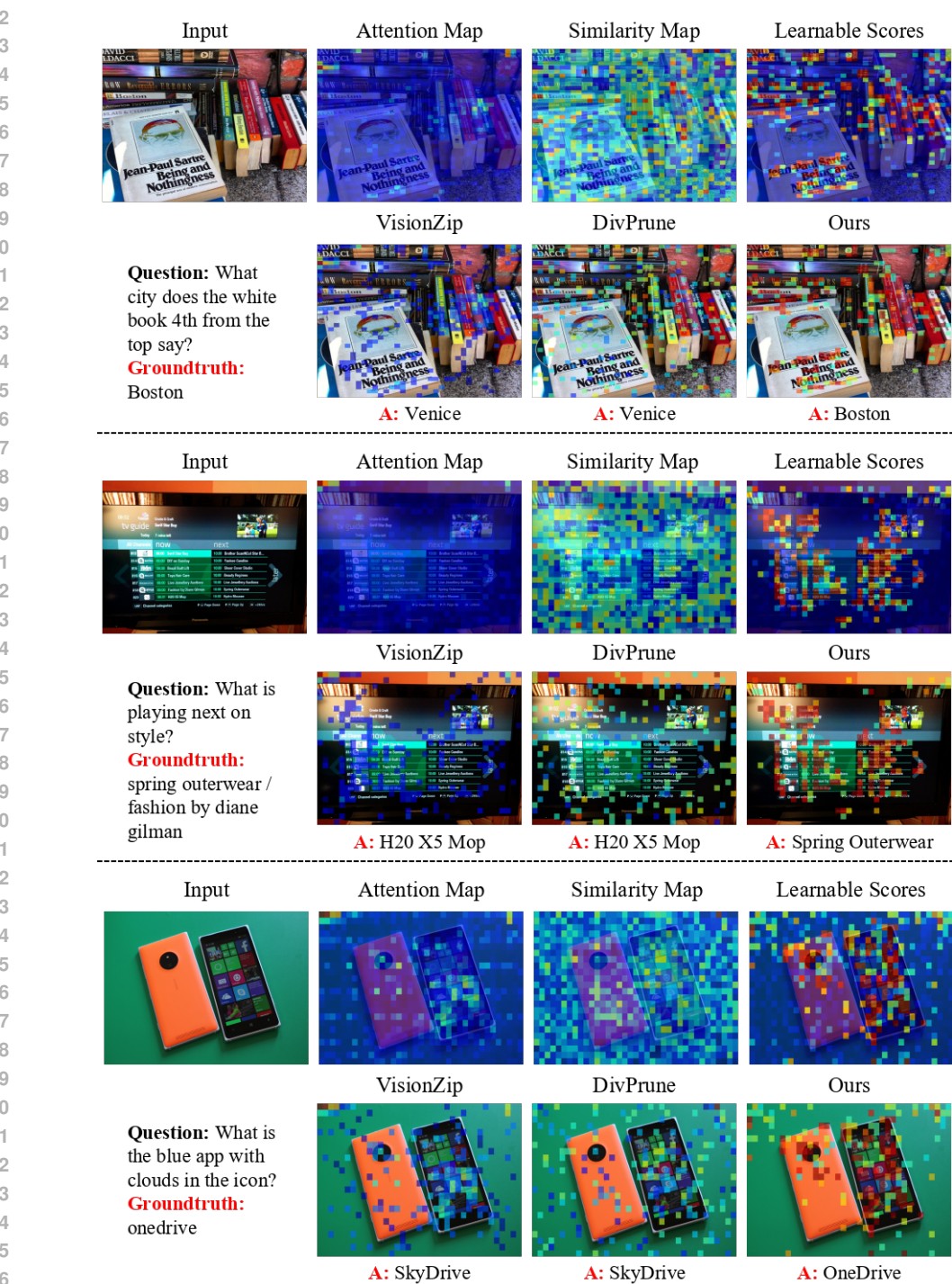

Figure 9: More qualitative comparison of VisionSelector on TextVQA. The top row visualizes different token scoring criteria: standard attention, a similarity map, and our learned importance scores. The bottom row shows the resulting token selections and model predictions at a 20% budget.

## A.12 USE OF LLMS

We use LLMs for grammar checking and text polishing to improve the readability of the paper.

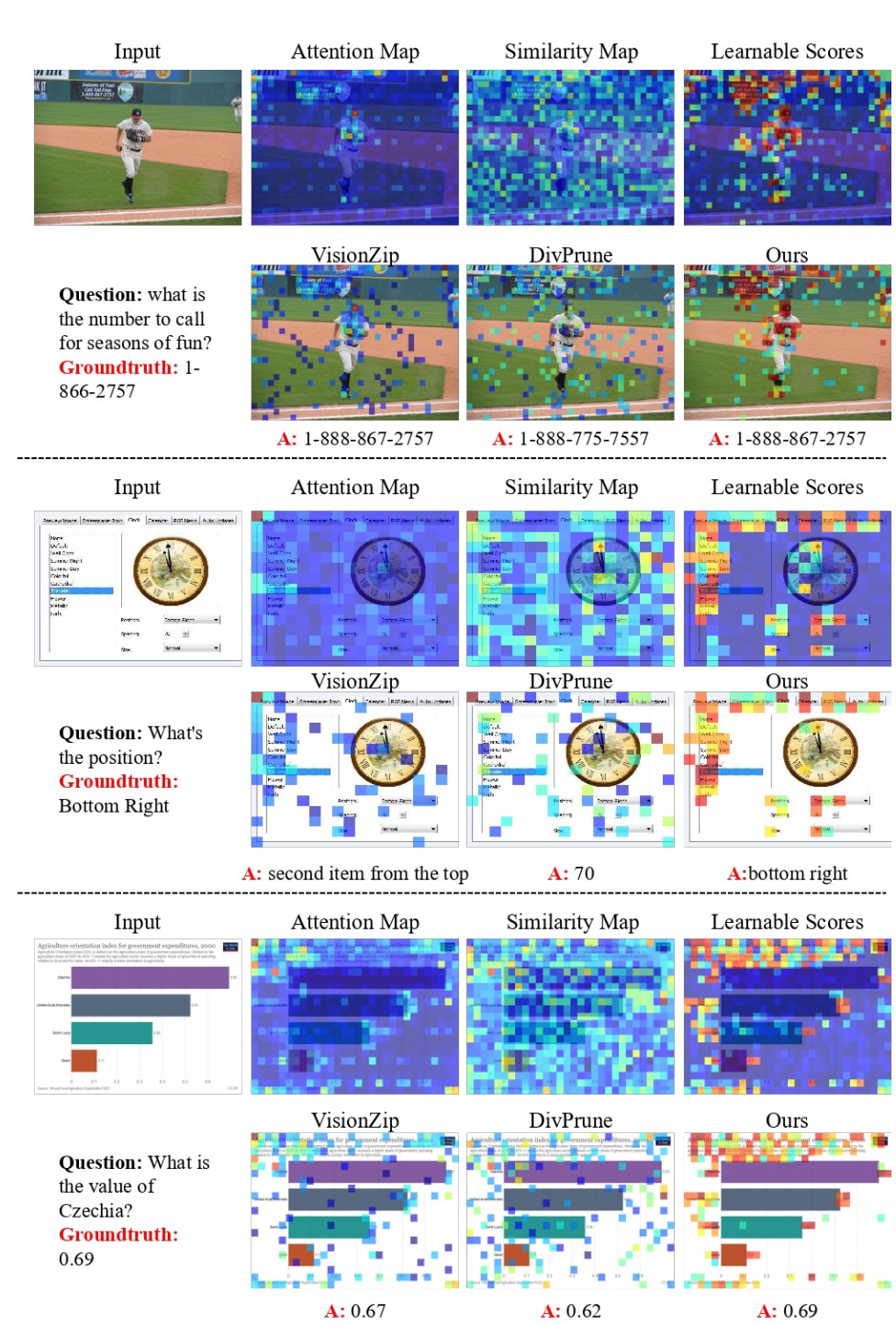

Figure 10: Qualitative comparison of VisionSelector on OCRBench. The top row visualizes different token scoring criteria: standard attention, a similarity map, and our learned importance scores. The bottom row shows the resulting token selections and model predictions at a 20% budget.

