# OpenReview forum: "VisionSelector: End-to-End Learnable Visual Token Compression for Efficient Multimodal LLMs"
_ICLR.cc/2026/Conference — Submitted to ICLR 2026_

### Official Review · Reviewer_8MrQ · 2025-10-28

**Soundness:** 3
**Presentation:** 2
**Contribution:** 2
**Rating:** 4
**Confidence:** 3

**Summary:**

This paper introduces VisionSelector, an end-to-end learnable visual token compression framework tailored for multimodal large language models (MLLMs). The key idea is to move beyond heuristic or fixed-rule post-processing of visual tokens, proposing instead a plug-and-play module that scores and selects tokens using a lightweight importance scorer, a differentiable Top-K mechanism, and a curriculum-based annealing strategy to bridge the gap between training and inference. VisionSelector is designed to be flexible, efficient, and easily integrated into existing MLLMs; importantly, it generalizes across different compression rates. Extensive experiments on diverse benchmarks—including image, video, and text-rich VQA datasets—demonstrate that VisionSelector achieves better accuracy-efficiency tradeoffs than prior state-of-the-art baselines, sometimes even improving performance beyond the uncompressed model by filtering noise.

**Strengths:**

- **Innovative End-to-End Framework:** The method reframes visual token pruning as a fully learnable, task-driven optimization process. By integrating the selection module as a lightweight and modular scorer, it sidesteps many brittleness and generalization issues that plague heuristic and rule-based approaches.

- **Differentiable Top-K with Curriculum Annealing:** Using a differentiable Top-K operator (see Section 3.3 and Equation 3, page 4–5) together with a curriculum annealing strategy (Section 3.4), VisionSelector provides a novel way of bridging the selection gap between soft selection during optimization and hard pruning at inference.

- **Plug-and-Play Integration:** The architecture (see Figure 2, page 3) is clean, minimally invasive, and compatible with existing MLLMs and inference accelerators (like FlashAttention).

**Weaknesses:**

1. **Missing and Incomplete Related Work Coverage:** The paper omits several relevant, recent works on token compression for multimodal LLMs (see the 'Potentially Missing Related Work' section for specifics), some of which could provide important baselines or complementary perspectives (e.g., LLaVA-Scissor, VoCo-LLaMA, TokenCarve), especially regarding adaptive or explainability-driven approaches. This literature gap is notable for a field moving as fast as MLLMs.
2. **Limited Theoretical Justification Beyond Empirical Performance:** While the differentiable Top-K mechanism and importance scorer are mechanically sound, theoretical analysis is minimal regarding optimality or generalization (e.g., what regimes guarantee retention of task-relevant tokens; how the annealing weight λ interacts with the main learning objective). There's no formal guarantee on information preservation aside from empirical robustness claims. Section 3.4, for instance, lacks even limited theoretical insight into the properties of the composite loss.
3. **Insufficient Discussion of Failure Modes and Bias:** The limitations section is brief and does not address corner cases or typical scenarios where hard selection could result in catastrophic output failures (e.g., if a batch of tokens is scored identically, or if rare-but-critical tokens are missed). It also does not analyze if some semantic categories are disproportionately pruned, which is a known failure in pruning/selection literature.
4. **Ablation Scope is Strong but Not Exhaustive:** Although ablations are included (see Table 3, 5), they mostly focus on data, λ annealing, and projection dimension. There's no targeted ablation on the scorer’s architecture (e.g., two layers vs. deeper/nonlinear networks, effects of near-zero initialization), or versions using only soft selection (removing hard-masking), which would clarify the necessity of each design choice.
5. **Results Reported Primarily on a Single MLLM Backbone:** The main text emphasizes results on Qwen2.5-VL-7B, with some reference to a 3B model in the appendix. However, the generalizability to other popular MLLM backbones (e.g., LLaVA, OpenFlamingo, GPT4-V, etc.) is not empirically validated, limiting claims of plug-and-play universality.

**Questions:**

Please see the weaknesses.

---

> ### Author Response · Authors · 2025-11-27
> **Response to Weakness 1 & Weakness 2**
>
> # Weakness 1
>
> > Missing and incomplete related work coverage.
>
> We thank the reviewer for the suggestion. We fully acknowledge the importance of recent works on token compression for MLLMs such as LLaVA-Scissor, VoCo-LLaMA, and TokenCarve, especially regarding their adaptive and explainability-driven approaches. We comprehensively add and analyze these references in **Related Work** of the revision to provide complementary perspectives. We also add discussions on LLaVA-mini and HoliTom. Furthermore we incorporate the learnable token selection scheme Dynamic-LLaVA as an experimental baseline. These additions allow us to present the context and contributions of our study in a more comprehensive and professional manner.
>
> ## Reference
>
> [1] LLaVA-Scissor: Token Compression with Semantic Connected Components for Video LLMs. Arxiv 2025.
>
> [2] Voco-llama: Towards vision compression with large language models. CVPR 2025.
>
> [3] Tokencarve: Information-preserving visual token compression in multimodal large language models. Arxiv 2025.
>
> [4] Llava-mini: Efficient image and video large multimodal models with one vision token. ICLR 2025.
>
> [5] HoliTom: Holistic Token Merging for Fast Video Large Language Models. NeurIPS 2025.
>
> [6] Dynamic-LLaVA: Efficient Multimodal Large Language Models via Dynamic Vision-language Context Sparsification. ICLR 2025.
>
> # Weakness 2
>
> > Limited theoretical justification beyond empirical performance.
>
> We thank the reviewer for the constructive comment. In **Appendix.9 (Theoretical Analysis of the Differentiable Top-K Operator)**, we explicitly analyze the conditions under which the DiffTopK operator favors task-relevant tokens. This analysis relies on three key properties including monotonicity, gradient stability with boundary maximization, and the competitive gradient dynamics induced by the cardinality constraint.
>
> **1.Order Consistency**
>
> The operator is strictly monotonic according to **Eq. (19)**, ensuring that the relative importance order is preserved. If the scorer correctly ranks a relevant token higher than noise, the DiffTopK operator faithfully reflects this in the selection probability $M$.
>
> **2.Boundary-Focused Optimization**
>
> Our Jacobian analysis in **Appendix.9.4** proves that gradient magnitudes are maximized when probabilities are near the decision boundary ($M\approx0.5$). This forces the optimization to aggressively resolve uncertainty for "hard-to-distinguish" tokens, driving them towards a definitive state (selected or rejected) rather than stagnating in the middle.
>
> **3.Competitive Gradient Dynamics**
>
> We analyze the backpropagation process in **Appendix 9.6** where the gradient calculation for each token incorporates a centering term derived from the weighted average of the group. This formulation establishes a competitive optimization environment. Specifically an unselected informative token with high gradient utility generates a corrective upward pressure that exceeds the background average and promotes its entry into the selected set. Simultaneously less relevant tokens experience downward pressure. This dynamic iteratively refines the subset by prioritizing features that significantly contribute to loss reduction.

---

> > ### Author Response · Authors · 2025-11-27
> > **Response to Weakness 3 & Weakness 4**
> >
> > # Weakness 3
> >
> > > Insufficient discussion of failure modes and bias.
> >
> > We thank the reviewer for the valuable comments on failure modes and bias. We broaden the discussion of these aspects in **Appendix.5** and **revise the Limitations section**.
> >
> > **1.Handling the corner case of identical scores**
> >
> > We clarify that exact equality of scores is practically impossible under floating point computation. Even in the theoretical regime where scores are very close, the differentiable Top K operator produces similar soft selection probabilities $M_i$. In this regime the entropy of the selection mask is high, and the constraint loss $L_{constraint}$ generates strong gradients during training that gradually drive these probabilities toward 0 or 1. This breaks the ambiguity and prevents the model from remaining in an unstable state where all tokens are effectively indistinguishable.
> >
> > **2.Rare but critical tokens**
> >
> > The visualizations in the **Figure 5** show that even under very low retention rates such as 2.5%, critical objects that occupy only a few pixels for example a skier in the snow still receive high importance scores and are preserved. When such high information density tokens are dropped, the task loss increases sharply and back propagation produces large gradients that force the LIS to increase their scores even if these tokens are rare. This indicates that VisionSelector is sensitive to information density rather than to raw pixel area.
> >
> > **3.Semantic pruning and failure case analysis**
> >
> > The qualitative visualizations in **Figure 5** and **Figure 6** suggest that the model does not exhibit systematic discrimination against specific semantic categories. However, the failure cases in **Figure 6** show that when an image contains multiple spatially scattered objects, an extreme token budget for example 10% may be insufficient to cover all semantics required by the image and the user query, which can lead to incorrect responses. This reveals a trade-off between compression rate and semantic completeness under extreme compression. In future work we plan to choose compression rates adaptively for different scenarios and to explore mechanisms for near lossless compression.
> >
> > # Weakness 4
> >
> > > Ablation scope is strong but not exhaustive.
> >
> > We thank the reviewer for the detailed suggestions regarding the ablation study. We add two targeted ablation experiments in the **Sec.4.6** and **Appendix.3** based on your advice.
> >
> > **1.Ablation on Scorer Initialization Strategy**
> >
> > We replace the near-zero initialization in the scorer architecture with Random initialization and Kaiming initialization. Experimental results (See **Table 7**) indicate that the overall performance of both variants across six benchmarks remains very close to the standard VisionSelector with near-zero initialization. This demonstrates that the scorer is not sensitive to specific initialization schemes.
> >
> > **2.Ablation on Removing Hard-Masking**
> >
> > We set the constraint loss weight $\lambda_t$ to 0 to remove hard-masking and utilize only soft selection. Results (See **Table 7**) show that performance drops when relying solely on soft selection. The incorporation of the hard-mask constraint further improves overall performance and enhances stability across different tasks.
> >
> > **Table 7: Ablation study on the impact of scorer initialization strategy, hard-masking constraint on model performance under Qwen2.5-VL-7B.**
> >
> > | Method                         | DocVQA | OCRBench | ScienceQA | MMMU  | MME     | POPE  | Avg    |
> > | ------------------------------ | ------ | -------- | --------- | ----- | ------- | ----- | ------ |
> > | Random Initialization          | 89.36  | 777      | 85.42     | 48.56 | 2311.13 | 84.92 | 96.36% |
> > | Kaiming Initialization         | 89.61  | 761      | 85.52     | 49.00 | 2311.41 | 84.83 | 96.23% |
> > | Remove Hard-Masking Constraint | 88.88  | 760      | 84.83     | 47.67 | 2284.06 | 83.43 | 95.05% |
> > | VisionSelector                 | 89.91  | 770      | 85.67     | 49.22 | 2293.54 | 84.27 | 96.33% |

---

> ### Author Response · Authors · 2025-11-27
> **Response to Weakness 5**
>
> # Weakness 5
>
> > Results Reported Primarily on a Single MLLM Backbone.
>
> We thank the reviewer for the suggestion regarding model universality. We conduct additional experiments on the recently released LLaVA-OneVision-1.5-8B [1] architecture in the **Appendix.8**. We compare VisionSelector with representative methods such as FastV, VisionZip, and DivPrune. Results (See **Table 8**) indicate that VisionSelector significantly outperforms existing methods on selected benchmark datasets across different compression rates. The substantial average performance improvement further verifies the plug-and-play universality of our method across different MLLM backbones.
>
> **Table 8: Comparison results of our method and different baselines on image-language understanding datasets under LLaVA-OneVision-1.5-8B. OOM denotes Out Of Memory.**
>
> | Method                       | DocVQA          | ChartQA         | TextVQA         | OCRBench      | ScienceQA       | A12D            | MMMU            | MME               | Avg              |
> | ---------------------------- | --------------- | --------------- | --------------- | ------------- | --------------- | --------------- | --------------- | ----------------- | ---------------- |
> | **Upper Bound (100%)** |                 |                 |                 |               |                 |                 |                 |                   |                  |
> | Avg. Visual Tokens           | 3795.45         | 595.92          | 978.56          | 830.65        | 320.4           | 523.27          | 670.11          | 1154.48           |                  |
> | LLaVA-OneVision-1.5-8B       | 97.87           | 86.72           | 79.51           | 830           | 98.56           | 94.01           | 56.44           | 2271.32           | 100%             |
> | **Retain 30% Tokens**  |                 |                 |                 |               |                 |                 |                 |                   |                  |
> | FastV (ECCV2024)             | 86.51           | 68.64           | 72.27           | 596           | 91.22           | 83.52           | 53.67           | 2019.84           | 86.96%           |
> | VisionZip (CVPR2025)         | oom             | 78.04           | 73.25           | oom           | 96.23           | 85.33           | oom             | oom               | /                |
> | DivPrune (CVPR2025)          | 85.23           | 70.64           | 76.03           | 645           | 94.65           | 89.35           | 54.11           | 2104.77           | 90.18%           |
> | VisionSelector               | **95.46** | **79.86** | **78.12** | **755** | **97.27** | **91.17** | **55.00** | **2241.82** | **96.39%** |
> | **Retain 20% Tokens**  |                 |                 |                 |               |                 |                 |                 |                   |                  |
> | FastV (ECCV2024)             | 76.64           | 59.08           | 67.33           | 489           | 87.95           | 78.89           | 51.33           | 1918.66           | 79.83%           |
> | VisionZip (CVPR2025)         | oom             | 67.56           | 66.28           | oom           | 91.87           | 79.18           | oom             | oom               | /                |
> | DivPrune (CVPR2025)          | 75.90           | 58.64           | 72.73           | 556           | 92.46           | 84.26           | 52.56           | 2057.83           | 83.85%           |
> | VisionSelector               | **90.51** | **74.72** | **76.02** | **677** | **95.79** | **90.45** | **55.11** | **2182.84** | **92.86%** |
> | **Retain 10% Tokens**  |                 |                 |                 |               |                 |                 |                 |                   |                  |
> | FastV (ECCV2024)             | 54.43           | 38.08           | 56.47           | 374           | 81.51           | 73.15           | 49.56           | 1800.00           | 67.89%           |
> | VisionZip (CVPR2025)         | oom             | 43.12           | 47.17           | oom           | 88.70           | 71.83           | oom             | oom               | /                |
> | DivPrune (CVPR2025)          | 56.48           | 38.24           | 65.48           | 419           | 88.20           | 76.85           | 51.33           | 1897.29           | 72.54%           |
> | VisionSelector               | **71.92** | **62.12** | **67.40** | **456** | **89.89** | **84.29** | **51.33** | **2002.44** | **80.60%** |
>
> ## Reference
>
> [1] LLaVA-OneVision-1.5: Fully Open Framework for Democratized Multimodal Training. Arxiv 2025.

---

### Official Review · Reviewer_zK6q · 2025-10-30

**Soundness:** 3
**Presentation:** 3
**Contribution:** 3
**Rating:** 4
**Confidence:** 4

**Summary:**

The paper addresses the explosion of visual tokens in Multimodal LLMs for high-res images, multi-image inputs, and video, which hurts memory and latency. Existing pruning/merging methods rely on heuristics (e.g., attention saliency) and often drop critical content under aggressive compression.

The authors propose VisionSelector, a lightweight (~12.85M params) plug-in module placed between the vision encoder and the LLM. It scores each visual token via a Learnable Importance Scorer (LIS), and uses a Differentiable Top-K Selection (DTS) scheme during training to learn which tokens to keep under a given budget; inference uses hard Top-K. The base MLLM remains frozen.

A Curriculum Annealing loss encourages the soft training mask to match the hard inference mask, aiming to close the train–inference gap.
Results claim: keeping only 10–30% of tokens still preserves near-baseline accuracy (sometimes ~100%), while reducing memory and roughly halving prefill latency. The same trained selector is said to generalize across different compression ratios and even to video benchmarks.

**Strengths:**

1. High Practicality
Performance retention remains exceptionally high (the paper claims ≈95% or even close to 100%) when retaining only a small number of visual tokens (10–30%). This approach reduces memory usage and nearly halves prefill inference latency. It offers significant real-world deployment value for high-resolution images, multi-page OCR, and video understanding.

2. Compact and pluggable
The backbone LLM requires no retraining whatsoever, training only a small scoring head with approximately 12.85 million parameters plus a Top-K module. This reduces training costs and facilitates direct deployment within existing multimodal large models.

**Weaknesses:**

1. LIS is essentially an attention-based scoring head; differentiable Top-K approximates Top-K as a continuous mask and then uses implicit differentiation; curriculum gradually increases consistency loss. These ideas closely resemble existing sparse selection/differentiable ranking/STE-style pruning. The paper packages them as a new paradigm but fails to clearly articulate what the truly novel theoretical contributions are.

2. Recommend comparing with methods that also require training to highlight the method's effectiveness.

3. It is recommended to add comparisons of more models, such as the llava series and internvl series.

4. Recommend adding a larger-scale model for comparison, such as the Qwen 2.5VL 14B.

**Questions:**

See the weaknesses section

---

> ### Author Response · Authors · 2025-11-27
> **Response to Weakness 1 (part 1/2)**
>
> # Weakness 1
>
> > Packages existing components (attention-scoring, differentiable Top-K, curriculum loss) as a new paradigm but fails to articulate novel theoretical contributions.
>
> We sincerely thank the reviewer for the comments regarding novelty. We address the concern that VisionSelector merely packages existing techniques like LIS, Differentiable Top-K, and CAS by adding comprehensive experiments in **Appendix.1** and revising the **Related Work** to clarify our core contribution. We construct a general end-to-end framework for the discrete optimization problem of MLLM visual token selection. This marks a paradigm shift from local independent gating to global differentiable ranking.
>
> **1.Motivation and Paradigm Shift**
>
> Heuristic rules risk attention sinking or pruning semantically dense tokens. Transformation-based or query-based compression methods are invasive and incur high training costs. Existing learnable selection methods such as Dynamic-LLaVA typically rely on independent token-wise gating based on local features. These approaches lack explicit modeling of relative token importance and fail to adapt to varying compression rates during inference.
>
> **2.Core Contribution and Component Synergy**
>
> To address these bottlenecks, we design a general end-to-end framework. The LIS serves as a lightweight scoring head that provides context-aware input for global ranking. The differentiable Top-K operator acts as a bridge between discrete selection and continuous optimization. It performs a soft relaxation on the ranking to enable gradient propagation to the LIS and allows end-to-end learning of relative token order. The Curriculum Annealing Strategy minimizes the discrepancy between soft and hard masks to ensure inference consistency.
>
> **3.Theoretical Analysis and Empirical Verification**
>
> We add proofs of monotonicity and shift invariance for our operator in **Appendix.9** of the revision to theoretically guarantee the preservation of relative importance order. We include comparison experiments with the SOTA learnable sparse selection method Dynamic-LLaVA, STE-style pruning method LightVLA, and differentiable ranking  operator SoftSort in **Appendix.1**. Results (See **Table 3**) show that our method outperforms existing learnable schemes.
>
> While our framework theoretically accommodates various differentiable sorting operators, we adopt the DiffTopK implementation to address critical bottlenecks in classical alternatives. Theoretically, literature such as SoftTopK (NeurIPS 2020) and DFTopK (Arxiv 2025) indicates that classical operators suffer from $O(N^2)$ complexity due to pairwise comparisons and face gradient conflicts arising from soft permutation matrix constraints. In contrast, our approach achieves $O(N)$ complexity and ensures gradient stability as proven in the **Appendix.9.3**. Empirically, addressing these limitations yields significant efficiency gains where our method reduces the training time of SoftSort from 70 minutes to 40 minutes. Furthermore, **Table 3** demonstrates that the DiffTopK-based VisionSelector consistently outperforms SoftSort-based baselines across all budgets. Consequently, we select this operator over classical alternatives for its superior efficiency and performance.
>
>
> Furthermore, Reviewer 8uak commends "The proposed end-to-end learnable token compression mechanism represents a major step forward in solving the problem of visual token inefficiency in MLLMs".
>
> **In summary**, VisionSelector is not merely a package of existing techniques but a customized framework for token compression. It successfully resolves the non-differentiability and inconsistency challenges in the transition from local thresholding to global ranking.

---

> > ### Author Response · Authors · 2025-11-27
> > **Response to Weakness 1 (part 2/2)**
> >
> > **Table 3: Comparison results of our method and three learnable alternatives on image-language understanding datasets under Qwen2.5-VL-7B.**
> >
> > | **Method**              | **DocVQA** | **ChatQA** | **TextVQA** | **OCRBench** | **ScienceQA** | **AI2D**  | **MMMU**  | **MME**     | **POPE**  | **Avg**    |
> > | ----------------------------- | ---------------- | ---------------- | ----------------- | ------------------ | ------------------- | --------------- | --------------- | ----------------- | --------------- | ---------------- |
> > | **Upper Bound (100%)**  |                  |                  |                   |                    |                     |                 |                 |                   |                 |                  |
> > | Qwen2.5-VL-7B                 | 94.33            | 83.40            | 82.84             | 838                | 87.26               | 93.59           | 50.78           | 2342.15           | 86.19           | 100%             |
> > | **Retain 30% Tokens**   |                  |                  |                   |                    |                     |                 |                 |                   |                 |                  |
> > | Dynamic (ICLR2025)            | 86.32            | 68.88            | 73.56             | 750                | 78.78               | 83.29           | 41.78           | 2180.42           | 80.87           | 88.99%           |
> > | STE-style Pruning (Arxiv2025) | 71.09            | 49.72            | 71.74             | 559                | 83.79               | 80.54           | 46.89           | 2204.46           | 84.28           | 83.85%           |
> > | SoftSort (ICML2020)           | 92.00            | 72.84            | 80.67             | 783                | 86.07               | 91.45           | 48.78           | 2311.28           | 84.05           | 96.03%           |
> > | VisionSelector                | **92.89**  | **72.96**  | **81.45**   | **809**      | **85.77**     | **92.00** | **50.11** | **2343.77** | **85.05** | **97.20%** |
> > | **Retain 20% Tokens**   |                  |                  |                   |                    |                     |                 |                 |                   |                 |                  |
> > | Dynamic (ICLR2025)            | 79.21            | 65.92            | 71.73             | 674                | 77.00               | 81.87           | 42.56           | 2117.37           | 77.22           | 85.51%           |
> > | STE-style Pruning (Arxiv2025) | 57.48            | 39.48            | 59.93             | 428                | 82.99               | 75.16           | 45.56           | 1985.39           | 81.71           | 75.16%           |
> > | SoftSort (ICML2020)           | 86.90            | 68.20            | 77.76             | 731                | 86.22               | 89.15           | 48.11           | 2259.01           | 82.77           | 92.92%           |
> > | VisionSelector                | **89.91**  | **68.84**  | **80.05**   | **770**      | **85.67**     | **90.15** | **49.22** | **2293.54** | **84.27** | **94.83%** |
> > | **Retain 10% Tokens**   |                  |                  |                   |                    |                     |                 |                 |                   |                 |                  |
> > | Dynamic (ICLR2025)            | 61.17            | 59.96            | 68.25             | 557                | 75.71               | 76.10           | 43.11           | 1977.58           | 70.80           | 78.35%           |
> > | STE-style Pruning (Arxiv2025) | 36.32            | 22.52            | 37.22             | 244                | 78.38               | 68.91           | 44.56           | 1744.56           | 75.55           | 61.43%           |
> > | SoftSort (ICML2020)           | 69.61            | 58.24            | 65.62             | 546                | 82.40               | 81.09           | 46.56           | 2014.89           | 78.09           | 81.93%           |
> > | VisionSelector                | **77.25**  | **62.28**  | **75.37**   | **646**      | **83.54**     | **84.72** | **47.44** | **2175.75** | **79.73** | **87.75%** |
> >
> > ## Reference
> >
> > [Dynamic-LLaVA] Dynamic-LLaVA: Efficient Multimodal Large Language Models via Dynamic Vision-language Context Sparsification. ICLR 2025.
> >
> > [LightVLA] The Better You Learn, The Smarter You Prune: Towards Efficient Vision-language-action Models via Differentiable Token Pruning. Arxiv 2025.
> >
> > [SoftSort] SoftSort: A Continuous Relaxation for the argsort Operator. ICML 2020.
> >
> > [SoftTopK] Differentiable Top-k with Optimal Transport. NeurIPS 2020.
> >
> > [DifferentiableTopK] Differentiable top-k classification learning. ICML 2022.
> >
> > [DFTopK] Differentiable Fast Top-K Selection for Large-Scale Recommendation. Arxiv 2025.

---

> > > ### Author Response · Authors · 2025-11-27
> > > **Response to Weakness 2**
> > >
> > > # Weakness 2
> > >
> > > > Recommend comparing with other trainable methods.
> > >
> > > We reimplement the Dynamic-LLaVA (ICLR2025,SoTA Methods) method on Qwen2.5-VL-7B (**Appendix.1**) and Qwen2.5-VL-3B (**Appendix.6**) using its official open-source code for a fair comparison. Both methods utilize plug-and-play token selection modules while Dynamic employs Gumbel Softmax token gates to generate discrete masks in attention layers. We strictly follow the official experimental settings and train only the token selection module. We will release the reimplementation code to support reproducibility. Experimental results (See **Table 4**) indicate that Dynamic performs well in some text-related tasks but suffers from significant performance degradation in general vision tasks. VisionSelector consistently outperforms Dynamic-LLaVA across different compression rates and maintains stable advantages in complex tasks such as DocVQA, OCRBench, and MME. This verifies the effectiveness of our method in token compression.
> > >
> > > **Table 4: Comparison results of our method and Dynamic on image-language understanding datasets under Qwen2.5-VL-7B.**
> > >
> > > | Method                       | DocVQA Anls     | ChatQA Relaxed  | TextVQA EM      | OCRBench EM   | ScienceQA Acc   | AI2D EM         | MMMU Acc        | MME Score         | POPE F1         | Avg              |
> > > | ---------------------------- | --------------- | --------------- | --------------- | ------------- | --------------- | --------------- | --------------- | ----------------- | --------------- | ---------------- |
> > > | **Upper Bound (100%)** |                 |                 |                 |               |                 |                 |                 |                   |                 |                  |
> > > | Qwen2.5-VL-7B                | 94.33           | 83.40           | 82.84           | 838           | 87.26           | 93.59           | 50.78           | 2342.15           | 86.19           | 100%             |
> > > | **Retain 30% Tokens**  |                 |                 |                 |               |                 |                 |                 |                   |                 |                  |
> > > | Dynamic (ICLR2025)           | 86.32           | 68.88           | 73.56           | 750           | 78.78           | 83.29           | 41.78           | 2180.42           | 80.87           | 88.99%           |
> > > | VisionSelector               | **92.89** | **72.96** | **81.45** | **809** | **85.77** | **92.00** | **50.11** | **2343.77** | **85.05** | **97.20%** |
> > > | **Retain 20% Tokens**  |                 |                 |                 |               |                 |                 |                 |                   |                 |                  |
> > > | Dynamic (ICLR2025)           | 79.21           | 65.92           | 71.73           | 674           | 77.00           | 81.87           | 42.56           | 2117.37           | 77.22           | 85.51%           |
> > > | VisionSelector               | **89.91** | **68.84** | **80.05** | **770** | **85.67** | **90.15** | **49.22** | **2293.54** | **84.27** | **94.83%** |
> > > | **Retain 10% Tokens**  |                 |                 |                 |               |                 |                 |                 |                   |                 |                  |
> > > | Dynamic (ICLR2025)           | 61.17           | 59.96           | 68.25           | 557           | 75.71           | 76.10           | 43.11           | 1977.58           | 70.80           | 78.35%           |
> > > | VisionSelector               | **77.25** | **62.28** | **75.37** | **646** | **83.54** | **84.72** | **47.44** | **2175.75** | **79.73** | **87.75%** |

---

> ### Author Response · Authors · 2025-11-27
> **Response to Weakness 3**
>
> # Weakness 3
>
> > Recommend adding comparisons of more models.
>
> We thank the reviewer for the valuable suggestion. We conduct additional experiments on the recently released LLaVA-OneVision-1.5-8B [1] model in the revision **Appendix.8**. We compare VisionSelector with representative methods such as FastV, VisionZip, and DivPrune. Results (See **Table 5**) indicate that VisionSelector significantly outperforms existing methods on selected benchmark datasets across different compression rates. The substantial average performance improvement further verifies the plug and play generality of our method across different MLLM backbones.
>
> **Table 5: Comparison results of our method and different baselines on image-language understanding datasets under LLaVA-OneVision-1.5-8B. OOM denotes Out Of Memory.**
>
> | Method                       | DocVQA          | ChartQA         | TextVQA         | OCRBench      | ScienceQA       | A12D            | MMMU            | MME               | Avg              |
> | ---------------------------- | --------------- | --------------- | --------------- | ------------- | --------------- | --------------- | --------------- | ----------------- | ---------------- |
> | **Upper Bound (100%)** |                 |                 |                 |               |                 |                 |                 |                   |                  |
> | Avg. Visual Tokens           | 3795.45         | 595.92          | 978.56          | 830.65        | 320.4           | 523.27          | 670.11          | 1154.48           |                  |
> | LLaVA-OneVision-1.5-8B       | 97.87           | 86.72           | 79.51           | 830           | 98.56           | 94.01           | 56.44           | 2271.32           | 100%             |
> | **Retain 30% Tokens**  |                 |                 |                 |               |                 |                 |                 |                   |                  |
> | FastV (ECCV2024)             | 86.51           | 68.64           | 72.27           | 596           | 91.22           | 83.52           | 53.67           | 2019.84           | 86.96%           |
> | VisionZip (CVPR2025)         | oom             | 78.04           | 73.25           | oom           | 96.23           | 85.33           | oom             | oom               | /                |
> | DivPrune (CVPR2025)          | 85.23           | 70.64           | 76.03           | 645           | 94.65           | 89.35           | 54.11           | 2104.77           | 90.18%           |
> | VisionSelector               | **95.46** | **79.86** | **78.12** | **755** | **97.27** | **91.17** | **55.00** | **2241.82** | **96.39%** |
> | **Retain 20% Tokens**  |                 |                 |                 |               |                 |                 |                 |                   |                  |
> | FastV (ECCV2024)             | 76.64           | 59.08           | 67.33           | 489           | 87.95           | 78.89           | 51.33           | 1918.66           | 79.83%           |
> | VisionZip (CVPR2025)         | oom             | 67.56           | 66.28           | oom           | 91.87           | 79.18           | oom             | oom               | /                |
> | DivPrune (CVPR2025)          | 75.90           | 58.64           | 72.73           | 556           | 92.46           | 84.26           | 52.56           | 2057.83           | 83.85%           |
> | VisionSelector               | **90.51** | **74.72** | **76.02** | **677** | **95.79** | **90.45** | **55.11** | **2182.84** | **92.86%** |
> | **Retain 10% Tokens**  |                 |                 |                 |               |                 |                 |                 |                   |                  |
> | FastV (ECCV2024)             | 54.43           | 38.08           | 56.47           | 374           | 81.51           | 73.15           | 49.56           | 1800.00           | 67.89%           |
> | VisionZip (CVPR2025)         | oom             | 43.12           | 47.17           | oom           | 88.70           | 71.83           | oom             | oom               | /                |
> | DivPrune (CVPR2025)          | 56.48           | 38.24           | 65.48           | 419           | 88.20           | 76.85           | 51.33           | 1897.29           | 72.54%           |
> | VisionSelector               | **71.92** | **62.12** | **67.40** | **456** | **89.89** | **84.29** | **51.33** | **2002.44** | **80.60%** |
>
> ## Reference
>
> [1] LLaVA-OneVision-1.5: Fully Open Framework for Democratized Multimodal Training. Arxiv 2025.

---

> ### Author Response · Authors · 2025-11-27
> **Response to Weakness 4**
>
> # Weakness 4
>
> > Recommend adding a larger-scale model for comparison.
>
> We thank the reviewer for the suggestion to include a larger-scale model for comparison. We evaluate VisionSelector on Qwen2.5-VL-32B in **Appendix.7** to verify its scalability. The results (See **Table 6**) indicate that our method maintains superior performance compared to other baselines. This validates the effectiveness of VisionSelector on large-scale models.
>
> **Table 6: Comparison results of our method and different baselines on image-language understanding datasets under Qwen2.5-VL-32B. OOM denotes Out Of Memory.**
>
> | Method                                              | DocVQA          | ChartQA         | TextVQA         | OCRBench      | ScienceQA       | AI2D            | MMMU            | MME               | Avg              |
> | --------------------------------------------------- | --------------- | --------------- | --------------- | ------------- | --------------- | --------------- | --------------- | ----------------- | ---------------- |
> | **Upper Bound (100%)**                        |                 |                 |                 |               |                 |                 |                 |                   |                  |
> | Avg. Visual Tokens                                  | 1951.61         | 596.06          | 976.58          | 652.82        | 323.05          | 510.19          | 601.15          | 867.67            |                  |
> | Qwen-2.5-VL-32B                                     | 91.10           | 64.20           | 66.76           | 649           | 91.62           | 94.20           | 60.11           | 2450.12           | 100%             |
> | **Retain 20% Tokens** |                 |                 |                 |               |                 |                 |                 |                   |                  |
> | FastV (ECCV2024)                                    | 54.89           | 36.24           | 53.18           | 371           | 81.80           | 82.41           | 55.78           | 1990.81           | 75.54%           |
> | VisionZip (CVPR2025)                                | 60.76           | 46.32           | 50.10           | 341           | 85.62           | 80.21           | _oom_         | 2094.90           | /                |
> | DART (EMNLP2025) | 51.22 | 35.24 | 57.91 | 443 | 85.37 | 74.29 | 56.22 | 2178.25 | 77.57% |
> | DivPrune (CVPR2025)                                 | 65.27           | 36.92           | 58.35           | 432           | 85.03           | 81.70           | 57.67           | **2208.52** | 81.09%           |
> | **VisionSelector**                            | **74.46** | **48.16** | **64.17** | **543** | **87.90** | **89.09** | **59.00** | 2197.50           | **89.36%** |
> | **Retain 10% Tokens** |                 |                 |                 |               |                 |                 |                 |                   |                  |
> | FastV (ECCV2024)                                    | 38.22           | 21.96           | 41.49           | 235           | 81.21           | 75.58           | 54.00           | 1752.91           | 63.10%           |
> | VisionZip (CVPR2025)                                | 35.76           | 28.80           | 36.38           | 242           | 82.80           | 71.5            | _oom_         | 1929.95           | /                |
> | DART (EMNLP2025) | 33.40 | 24.88 | 47.83 | 342 | 81.06 | 68.52 | 54.67 | 2029.26 | 66.84% |
> | DivPrune (CVPR2025)                                 | 48.01           | 25.40           | 51.00           | 310           | 81.31           | 71.92           | 54.78           | **2072.38** | 69.65%           |
> | **VisionSelector**                            | **53.1**  | **41.96** | **59.66** | **413** | **83.69** | **81.02** | **55.33** | 2007.73           | **78.50%** |

---

### Official Review · Reviewer_8uak · 2025-11-02

**Soundness:** 3
**Presentation:** 3
**Contribution:** 3
**Rating:** 6
**Confidence:** 4

**Summary:**

This paper introduces a new framework for adaptive visual token compression in MLLMs. The authors identify a key inefficiency in current MLLMs that the excessive number of visual tokens from high-resolution or multi-image inputs, and argue that existing heuristic or attention-based compression methods either discard critical information or suffer from “attention sink” biases.
To address this, they propose VisionSelector, a lightweight, plug-and-play module that reformulates token compression as a fully learnable decision process. It comprises a Learnable Importance Scorer, a Differentiable Top-K selection mechanism, and a Curriculum Annealing Strategy that bridges the training–inference gap. Trained once at a fixed compression rate, VisionSelector generalizes to arbitrary retention budgets and integrates seamlessly with MLLMs such as Qwen2.5-VL. Extensive experiments across 13 image and video understanding benchmarks demonstrate that VisionSelector achieves superior accuracy retention and substantially outperforming prior methods.

**Strengths:**

1. The proposed end-to-end learnable token compression mechanism represents a major step forward in solving the problem of visual token inefficiency in MLLMs. It offers a dynamic, adaptive solution as opposed to rigid, heuristic-based methods.
2. The approach is highly practical. The module is lightweight with only 12.85M params, plug-and-play, and fast to train as it keeps the backbone frozen.
3. The method is designed to integrate with existing MLLMs without requiring changes to the backbone model, providing a practical and deployable solution for current systems.

**Weaknesses:**

1. While the paper motivates limitations of attention/similarity heuristics, it does not compare to strong learned pruning/gating baselines (e.g., Gumbel‑Softmax token gates, NeuralSort/SoftSort‑style Top‑K relaxations, recent dynamic‑context sparsification for MLLMs). The DiffTop‑K reference is a StackExchange post rather than established differentiable sorting literature. A stronger related‑work discussion and head‑to‑head comparisons to learnable alternatives would clarify novelty and technical significance.

2. All experiments are conducted on the Qwen2.5-VL model (both 7B and 3B variants). Given the claim of “plug‑and‑play” generality, the evidence would be stronger with other popular MLLMs (e.g., InternVL, LLaVA, etc.), especially those without PatchMerger modules. Testing VisionSelector on at least one other major architectural family is necessary to fully validate this claim of general applicability.

3. The ablation study in Table 3. reveals that the model's performance is highly sensitive to the training data mix. For instance, adding the COCO dataset significantly boosts performance on OCR-related tasks (OCRBench score from 701 to 763). This highlights a key trade-off that the paper glosses over: while this learnable method is more powerful, it is also data-dependent and requires careful data curation (144K samples) and training.

**Questions:**

While VisionSelector performs well in training with soft selection, does the transition to hard selection during inference introduce any discrepancies, particularly in edge cases with low compression budgets?

**Details Of Ethics Concerns:**

None.

---

> ### Author Response · Authors · 2025-11-27
> **Response to Weakness 1 (part 1/3)**
>
> # Weakness 1
>
> > Lacks comparison to learned baselines. The DiffTop‑K relies on non-academic reference. Requires stronger baselines to demonstrate novelty.
>
> **Response:**
>
> We thank the reviewer for suggesting the inclusion of learning-based baselines and a stronger discussion of related work. We add comprehensive experiments in **Appendix.1** and revise the **Related Work**. The details follow below.
>
> **Table 1: Comparison results of our method and three learnable alternatives on image-language understanding datasets under Qwen2.5-VL-7B.**
>
> | **Method**              | **DocVQA** | **ChatQA** | **TextVQA** | **OCRBench** | **ScienceQA** | **AI2D**  | **MMMU**  | **MME**     | **POPE**  | **Avg**    |
> | ----------------------------- | ---------------- | ---------------- | ----------------- | ------------------ | ------------------- | --------------- | --------------- | ----------------- | --------------- | ---------------- |
> | **Upper Bound (100%)**  |                  |                  |                   |                    |                     |                 |                 |                   |                 |                  |
> | Qwen2.5-VL-7B                 | 94.33            | 83.40            | 82.84             | 838                | 87.26               | 93.59           | 50.78           | 2342.15           | 86.19           | 100%             |
> | **Retain 30% Tokens**   |                  |                  |                   |                    |                     |                 |                 |                   |                 |                  |
> | Dynamic (ICLR2025)            | 86.32            | 68.88            | 73.56             | 750                | 78.78               | 83.29           | 41.78           | 2180.42           | 80.87           | 88.99%           |
> | STE-style Pruning (Arxiv2025) | 71.09            | 49.72            | 71.74             | 559                | 83.79               | 80.54           | 46.89           | 2204.46           | 84.28           | 83.85%           |
> | SoftSort (ICML2020)           | 92.00            | 72.84            | 80.67             | 783                | 86.07               | 91.45           | 48.78           | 2311.28           | 84.05           | 96.03%           |
> | VisionSelector                | **92.89**  | **72.96**  | **81.45**   | **809**      | **85.77**     | **92.00** | **50.11** | **2343.77** | **85.05** | **97.20%** |
> | **Retain 20% Tokens**   |                  |                  |                   |                    |                     |                 |                 |                   |                 |                  |
> | Dynamic (ICLR2025)            | 79.21            | 65.92            | 71.73             | 674                | 77.00               | 81.87           | 42.56           | 2117.37           | 77.22           | 85.51%           |
> | STE-style Pruning (Arxiv2025) | 57.48            | 39.48            | 59.93             | 428                | 82.99               | 75.16           | 45.56           | 1985.39           | 81.71           | 75.16%           |
> | SoftSort (ICML2020)            | 86.90            | 68.20            | 77.76             | 731                | 86.22               | 89.15           | 48.11           | 2259.01           | 82.77           | 92.92%           |
> | VisionSelector                | **89.91**  | **68.84**  | **80.05**   | **770**      | **85.67**     | **90.15** | **49.22** | **2293.54** | **84.27** | **94.83%** |
> | **Retain 10% Tokens**   |                  |                  |                   |                    |                     |                 |                 |                   |                 |                  |
> | Dynamic (ICLR2025)            | 61.17            | 59.96            | 68.25             | 557                | 75.71               | 76.10           | 43.11           | 1977.58           | 70.80           | 78.35%           |
> | STE-style Pruning (Arxiv2025) | 36.32            | 22.52            | 37.22             | 244                | 78.38               | 68.91           | 44.56           | 1744.56           | 75.55           | 61.43%           |
> | SoftSort (ICML2020)           | 69.61            | 58.24            | 65.62             | 546                | 82.40               | 81.09           | 46.56           | 2014.89           | 78.09           | 81.93%           |
> | VisionSelector                | **77.25**  | **62.28**  | **75.37**   | **646**      | **83.54**     | **84.72** | **47.44** | **2175.75** | **79.73** | **87.75%** |

---

> ### Author Response · Authors · 2025-11-27
> **Response to Weakness 1 (part 2/3)**
>
> **1. Comparison with SOTA Learnable Baseline Dynamic-LLaVA (ICLR 2025)**
>
> We reimplement the Dynamic-LLaVA method on Qwen2.5-VL-7B using its official open-source code for a fair comparison (See **Table 1**). Both methods utilize plug-and-play token selection modules while Dynamic-LLaVA employs Gumbel Softmax token gates to generate discrete masks in attention layers. We strictly follow the official experimental settings and train only the token selection module. We plan to release the reimplementation code to support reproducibility. Experimental results indicate that Dynamic-LLaVA performs well in some text-related tasks but suffers from significant performance degradation in general vision tasks. VisionSelector consistently outperforms Dynamic-LLaVA across different compression rates and maintains stable advantages in complex tasks such as DocVQA and MME. This verifies the effectiveness of our method.
>
> **2. Comparison with Recent Dynamic‑context Sparsification for MLLMs (Arxiv 2025)**
>
> We acknowledge the recent dynamic-context sparsification for MLLMs named LightVLA (Arxiv 2025). To conduct a fair head-to-head comparison, we replace the differentiable Top-K operator in VisionSelector with the STE-style pruning strategy employed by LightVLA while maintaining all other experimental settings.
>
> The suboptimal results (See **Table 1**) of STE-style pruning likely stem from the absence of global sorting and the STE's inherent gradient mismatch issue pointed Gumbel-Sofmax. Conversely, our DiffTopK approach ensures theoretically guaranteed gradient stability and monotonicity, as demonstrated in DFTopK and **Appendix.9.3**, enabling more effective end-to-end optimization.
>
> **3. Comparison with Classical Differentiable Sorting Operator SoftSort (ICML 2020)**
>
> We replace DiffTopK in VisionSelector with SoftSort while keeping other settings unchanged. Results (See **Table 1**) show that the SoftSort-based VisionSelector significantly outperforms comparative methods including Dynamic-LLaVA, VisionZip, and DivPrune. This demonstrates that our end-to-end framework achieves operator decoupling and replaceability. Furthermore, the DiffTopK-based VisionSelector performs better than the SoftSort-based variant and validates the effectiveness of the DiffTopK operator.
>
> **4.Theoretical Citations and Technical Significance of DiffTopK**
>
> We add citations for differentiable sorting methods like **NeuralSort (ICLR2020)** and **SoftSort (ICML2020)** in the **Sec.2.2**. We also add references for differentiable TopK operators including **SoftTopK (NeurIPS2020)**, **DifferentiableTopK (ICML2022)**, **DSTopK (ICML2023)**, and the recent **DFTopK (Arxiv2025)** in **Sec.3.3**. We deeply discuss the distinctions between VisionSelector and existing learnable token selection schemes like Dynamic-LLaVA in **Sec.2.1**. Existing schemes rely on independent token-wise gating based on local features and fail to model the relative importance among tokens. We design a general end-to-end framework that shifts the paradigm from local independent decisions to global differentiable sorting decisions. We introduce DiffTopK to enable global context awareness. This ensures that token retention relies on ranking superiority across the full sequence rather than isolated token decisions.
>
> While our framework theoretically accommodates various differentiable sorting operators, we adopt the DiffTopK implementation to address critical bottlenecks in classical alternatives. Theoretically, literature such as SoftTopK and DFTopK indicates that classical operators suffer from $O(N^2)$ complexity due to pairwise comparisons and face gradient conflicts arising from soft permutation matrix constraints. In contrast, our approach achieves $O(N)$ complexity and ensures gradient stability as proven in the **Appendix.9.3**. Empirically, addressing these limitations yields significant efficiency gains where our method reduces the training time of SoftSort from 70 minutes to 40 minutes. Furthermore, **Table 1** demonstrates that the DiffTopK-based VisionSelector consistently outperforms SoftSort-based baselines across all budgets. Consequently, we select this operator over classical alternatives for its superior efficiency and performance.

---

> > ### Author Response · Authors · 2025-11-27
> > **Response to Weakness 1 (part 3/3)**
> >
> > ## Reference
> >
> > [Dynamic-LLaVA] Dynamic-LLaVA: Efficient Multimodal Large Language Models via Dynamic Vision-language Context Sparsification. ICLR 2025.
> >
> > [Gumbel-Softmax] Categorical Reparameterization with Gumbel-Softmax. ICLR 2017.
> >
> > [LightVLA] The Better You Learn, The Smarter You Prune: Towards Efficient Vision-language-action Models via Differentiable Token Pruning. Arxiv 2025.
> >
> > [STE] Estimating or propagating gradients through stochastic neurons for conditional computation. Arxiv 2013.
> >
> > [NeuralSort] Stochastic Optimization of Sorting Networks via Continuous Relaxations. ICLR 2020.
> >
> > [SoftSort] SoftSort: A Continuous Relaxation for the argsort Operator. ICML 2020.
> >
> > [SoftTopK] Differentiable Top-k with Optimal Transport. NeurIPS 2020.
> >
> > [DifferentiableTopK] Differentiable top-k classification learning. ICML 2022.
> >
> > [DSTopK] Fast, differentiable and sparse top-k: a convex analysis perspective. ICML 2023.
> >
> > [DFTopK] Differentiable Fast Top-K Selection for Large-Scale Recommendation. Arxiv 2025.

---

> ### Author Response · Authors · 2025-11-27
> **Response to Weakness 2**
>
> # Weakness 2
>
> > Experiments limited to Qwen2.5-VL; lacks testing on diverse MLLMs to validate plug-and-play generality.
>
> We thank the reviewer for the suggestion regarding model generality. We conduct additional experiments on the recently released LLaVA-OneVision-1.5-8B [1] architecture in the **Appendix.8**. We compare VisionSelector with representative methods such as FastV, VisionZip, and DivPrune. Results (See **Table 2**) indicate that VisionSelector significantly outperforms existing methods on selected benchmark datasets across different compression rates. The substantial average performance improvement further verifies the plug-and-play generality of our method across different MLLM backbones.
>
> **Table 2: Comparison results of our method and different baselines on image-language understanding datasets under LLaVA-OneVision-1.5-8B. OOM denotes Out Of Memory.**
>
> | Method                       | DocVQA          | ChartQA         | TextVQA         | OCRBench      | ScienceQA       | A12D            | MMMU            | MME               | Avg              |
> | ---------------------------- | --------------- | --------------- | --------------- | ------------- | --------------- | --------------- | --------------- | ----------------- | ---------------- |
> | **Upper Bound (100%)** |                 |                 |                 |               |                 |                 |                 |                   |                  |
> | Avg. Visual Tokens           | 3795.45         | 595.92          | 978.56          | 830.65        | 320.4           | 523.27          | 670.11          | 1154.48           |                  |
> | LLaVA-OneVision-1.5-8B       | 97.87           | 86.72           | 79.51           | 830           | 98.56           | 94.01           | 56.44           | 2271.32           | 100%             |
> | **Retain 30% Tokens**  |                 |                 |                 |               |                 |                 |                 |                   |                  |
> | FastV (ECCV2024)             | 86.51           | 68.64           | 72.27           | 596           | 91.22           | 83.52           | 53.67           | 2019.84           | 86.96%           |
> | VisionZip (CVPR2025)         | oom             | 78.04           | 73.25           | oom           | 96.23           | 85.33           | oom             | oom               | /                |
> | DivPrune (CVPR2025)          | 85.23           | 70.64           | 76.03           | 645           | 94.65           | 89.35           | 54.11           | 2104.77           | 90.18%           |
> | VisionSelector               | **95.46** | **79.86** | **78.12** | **755** | **97.27** | **91.17** | **55.00** | **2241.82** | **96.39%** |
> | **Retain 20% Tokens**  |                 |                 |                 |               |                 |                 |                 |                   |                  |
> | FastV (ECCV2024)             | 76.64           | 59.08           | 67.33           | 489           | 87.95           | 78.89           | 51.33           | 1918.66           | 79.83%           |
> | VisionZip (CVPR2025)         | oom             | 67.56           | 66.28           | oom           | 91.87           | 79.18           | oom             | oom               | /                |
> | DivPrune (CVPR2025)          | 75.90           | 58.64           | 72.73           | 556           | 92.46           | 84.26           | 52.56           | 2057.83           | 83.85%           |
> | VisionSelector               | **90.51** | **74.72** | **76.02** | **677** | **95.79** | **90.45** | **55.11** | **2182.84** | **92.86%** |
> | **Retain 10% Tokens**  |                 |                 |                 |               |                 |                 |                 |                   |                  |
> | FastV (ECCV2024)             | 54.43           | 38.08           | 56.47           | 374           | 81.51           | 73.15           | 49.56           | 1800.00           | 67.89%           |
> | VisionZip (CVPR2025)         | oom             | 43.12           | 47.17           | oom           | 88.70           | 71.83           | oom             | oom               | /                |
> | DivPrune (CVPR2025)          | 56.48           | 38.24           | 65.48           | 419           | 88.20           | 76.85           | 51.33           | 1897.29           | 72.54%           |
> | VisionSelector               | **71.92** | **62.12** | **67.40** | **456** | **89.89** | **84.29** | **51.33** | **2002.44** | **80.60%** |
>
> ## Reference
>
> [1] LLaVA-OneVision-1.5: Fully Open Framework for Democratized Multimodal Training. Arxiv 2025.

---

> ### Author Response · Authors · 2025-11-27
> **Response to Weakness 3 & Question 1**
>
> # Weakness 3
>
> > Performance highly data-sensitive.
>
> We thank the reviewer for the discussion regarding data sensitivity and curation costs. We appreciate the opportunity to clarify this trade-off from three perspectives.
>
> Specifically, we observe that VisionSelector maintains superior performance even **without the COCO dataset (Config-2 in Table 5)**. It significantly outperforms the second-best methods across multiple benchmarks, achieving 87.59 versus 75.99 on DocVQA, 701 versus 648 on OCRBench, and 2293.54 versus 2219.3 on MME. This suggests that the core effectiveness of our method is robust and not solely dependent on specific data mixtures.
>
> Furthermore, the performance boost from adding COCO likely reflects our model's capacity to benefit from diverse data distributions rather than a strict dependency. This indicates that VisionSelector scales effectively when provided with varied training signals.
>
> Regarding the data source, the 144K training samples do not require specialized curation. They represent a standard subset of open-source multimodal instruction tuning datasets, aligning with common practices in LLaVA [1], Dynamic-LLaVA [2], Cambrian-1 [3], and LLaVA-OneVision [4]. Consequently, our approach integrates seamlessly into existing training pipelines without necessitating extra manual cleaning or complex data construction.
>
> ## Reference
>
> [1] Visual Instruction Tuning. NeurIPS 2023.
>
> [2] Dynamic-LLaVA: Efficient Multimodal Large Language Models via Dynamic Vision-language Context Sparsification. ICLR 2025.
>
> [3] Cambrian-1: A Fully Open, Vision-Centric Exploration of Multimodal LLMs. NeurIPS 2024.
>
> [4] LLaVA-OneVision: Easy Visual Task Transfer. Arxiv 2024.
>
> # Question 1
>
> > Inference discrepancy from soft-to-hard selection.
>
> We thank the reviewer for the concern regarding the potential discrepancy between soft selection during training and hard selection during inference. We acknowledge that differences exist due to this transition. To mitigate this issue, we employ a constraint loss $\lambda_{constraint}$ and a Curriculum Annealing Strategy. These mechanisms progressively bridge the gap between training and inference modes. The soft masks converge toward binary hard masks as the model converges. Visualizations in **Figure 5** and **Figure 6** of the revision show that the weights of selected important tokens concentrate around 1 (red) which indicates a state close to hard selection. Furthermore, experimental results validate the robustness of our approach. VisionSelector maintains superior performance compared to other methods on models including the Qwen2.5-VL series and LLaVA-OneVision-1.5-8B even in edge cases with extreme compression budgets where we retain only 10% of tokens.

---

### Author Response · Authors · 2025-11-27
**General Responses, Contributions Summary and Revision Summary**

Dear Area Chair and reviewers,

We sincerely thank all the reviewers for their constructive suggestion and valuable reviews. We are especially grateful for the recognition of our novel end-to-end learnable framework as a major step forward from heuristic methods (Reviewers 8uak, 8MrQ), the high practicality and exceptional performance retention even under high compression rates (Reviewer zK6q), the lightweight, plug-and-play design that requires no backbone retraining (Reviewers 8uak, zK6q, 8MrQ), the significant real-world value for high-resolution and video understanding tasks (Reviewer zK6q), and the technical innovation of the Differentiable Top-K operator combined with curriculum annealing (Reviewer 8MrQ).

**We hereby reiterate our novelty and contributions.**

The computational and memory bottlenecks caused by massive visual tokens are major pain points in current MLLM inference. Existing token compression methods have significant limitations. Heuristic-based methods are prone to attention sinking or losing critical information. Transformation-based or query-based methods are invasive as they alter the original feature distribution and require high retraining costs. Existing learnable token selection methods typically rely on independent token-wise gating. This decision process based on local thresholds lacks explicit modeling of the relative importance among global tokens and fails to adapt flexibly to different compression rates during inference. To address the discrete optimization challenge of MLLM visual token selection, **we construct a general end-to-end learnable framework. This marks a paradigm shift from local independent gating to global differentiable ranking**. VisionSelector incorporates a Learnable Importance Scorer (LIS) to explicitly model global ranking. A Differentiable Top-K Operator solves the gradient back-propagation issue in discrete selection through continuous relaxation while preserving the relative importance order. Additionally, the Curriculum Annealing Strategy (CAS) reduces the inconsistency between training and inference. VisionSelector is cost-efficient with only 12.85MB of trainable parameters and requires approximately 40 minutes of training on 8 NVIDIA A800 GPUs. We evaluate VisionSelector on Qwen2.5-VL-3B/7B/32B and LLaVA-OneVision-1.5-8B across three compression rates. Experimental results demonstrate that our method outperforms existing approaches in performance, inference speed and memory cost, which proves the plug-and-play generality of our method.

We appreciate your thoughtful questions and concerns, and we have tried our best to address them comprehensively in our individual responses.

We have carefully revised the paper to reflect our responses to reviewers' concerns and suggestions (**highlighted in blue**). Key revisions are:

* Revise the **Related Work** section to include recent studies and clarify the distinctions between our method, existing learnable approaches, and differentiable sorting operators.
* Add experiments on LLaVA-OneVision-1.5-8B in **Appendix.8** to verify the plug-and-play generality of our method.
* Add comparative experiments in **Appendix.1** and **Appendix.6** against Dynamic-LLaVA on Qwen2.5-VL-3B/7B.
* Add comparative experiments in **Appendix.1** against STE-style pruning, and SoftSort.
* Add theoretical analyses for the differentiable Top-K operator in **Appendix.9** to verify its order-preserving property and gradient stability.
* Add performance experiments on Qwen2.5-VL-32B in **Appendix.7** to verify the scalability.
* Add the discussion on failure modes under extreme compression scenarios in **Appendix.5** and add more discussions on limitations in the Limitations section.
* Add ablation studies in **Sec.4.6** and **Appendix.3** regarding LIS initialization and hard-masking constraints.
* Update ScienceQA results using Lmms-Eval v0.5 to account for evaluation discrepancies with the previously used v0.3 and ensure fair alignment with LLaVA-OneVision-1.5-8B.

Thanks, Authors

---

### Comment · Area_Chair_tsyK · 2025-11-28

Dear Reviewers, the discussion period is about to close. We kindly ask you to participate in the discussion or update your score based on the authors' rebuttal before the deadline. Thank you for your time and valuable contribution!

---

### Meta-Review · Area_Chair_SL2Y · 2025-12-03

**Summary:**

This paper proposes VisionSelector, an end-to-end learnable framework for compressing visual tokens in Multimodal LLMs to improve efficiency. It employs a lightweight scorer module with a differentiable Top-K operator and a curriculum annealing strategy to select important tokens adaptively, claiming strong performance retention under high compression rates while being plug-and-play. The three reviewers assigned scores of 6, 4, and 4. They raised several shared and significant concerns. The primary issues included a lack of comparison with strong learnable baseline methods (e.g., Dynamic-LLaVA, SoftSort), insufficient evidence for the claimed "plug-and-play" generality across diverse MLLM architectures beyond Qwen2.5-VL, and questions about the core technical novelty, as the framework was seen as a combination of existing components like attention-based scoring and differentiable sorting. Additional concerns covered limited theoretical grounding, an incomplete discussion of failure modes and biases, and an insufficient ablation study scope. In the rebuttal, the authors provided extensive new experiments. They added comparisons against Dynamic-LLaVA, STE-style pruning, and SoftSort, showing superior results and so on.  Before the rolling back of the OpenReview system, no reviewers give positive response.

This paper is finally not recommended for acceptance because of the following reasons. Firstly, as pointed out by reviewers, the original submission of this paper has some clear problems, such as insufficient comparison. Although authors have added new results during rebuttal, I can not guarantee that these results fully solved the concerns of reviewers since none of them gave new response. Second, compared with other papers in my AC batch, the rating of this paper is lower than the average rating of accepted papers. And I do not find a very strong reason to support acceptance. I am recommending a rejection, but I wouldn't mind if the paper gets accepted.

**Reviewer Concerns:**

The authors' rebuttal adequately addressed specific experimental requests, such as adding comparisons with learnable baselines and testing on more model backbones. However, the more fundamental concerns regarding the perceived lack of significant technical novelty and the depth of theoretical contribution remain outstanding and were central to the reviewers' original reservations.

**Reviewer Scores:**

Given the comprehensive additional experiments provided in the rebuttal, it is possible that the reviewers might have slightly increased their scores, particularly regarding the aspects of generality and baseline comparison.

---

### Decision · Program_Chairs · 2026-01-26

Reject